# The E3 ubiquitin ligase Pellino2 mediates priming of the NLRP3 inflammasome

Fiachra Humphries[1], Ronan Bergin[1], Ruaidhri Jackson[1], Nezira Delagic[1], Bingwei Wang[1], Shuo Yang[1,2], Alice V. Dubois[3], Rebecca J. Ingram[3] & Paul N. Moynagh[1,3]

The NLRP3 inflammasome has an important function in inflammation by promoting the processing of pro-IL-1β and pro-IL-18 to their mature bioactive forms, and by inducing cell death via pyroptosis. Here we show a critical function of the E3 ubiquitin ligase Pellino2 in facilitating activation of the NLRP3 inflammasome. Pellino2-deficient mice and myeloid cells have impaired activation of NLRP3 in response to toll-like receptor priming, NLRP3 stimuli and bacterial challenge. These functions of Pellino2 in the NLRP3 pathway are dependent on Pellino2 FHA and RING-like domains, with Pellino2 promoting the ubiquitination of NLRP3 during the priming phase of activation. We also identify a negative function of IRAK1 in the NLRP3 inflammasome, and describe a counter-regulatory relationship between IRAK1 and Pellino2. Our findings reveal a Pellino2-mediated regulatory signaling system that controls activation of the NLRP3 inflammasome.

[1] Institute of Immunology, Department of Biology, National University of Ireland Maynooth, Maynooth, Co. Kildare, Ireland. [2] Department of Immunology, Nanjing Medical University, Nanjing, Jiangsu Province, China. [3] Centre for Experimental Medicine, Queen's University Belfast, Belfast, UK. These authors contributed equally: Fiachra Humphries, Ronan Bergin. Correspondence and requests for materials should be addressed to P.N.M. (email: Paul.Moynagh@mu.ie)

Pattern recognition receptors (PRR) are utilized by the innate immune system to recognize pathogen-associated molecular patterns (PAMP) and trigger the induction of pro-inflammatory cytokines that will promote pathogen killing and removal from the host[1]. Transmembrane Toll-like receptors (TLR) and cytosolic nucleotide-binding and oligomerization domain (NOD)-like receptors (NLR) are important families of PRRs[2]. TLRs have a cytoplasmic Toll/interleukin-1(IL-1) receptor (TIR) domain and most TLRs, upon ligand engagement, interact with the intracellular TIR domain-containing adapter myeloid differentiation primary response protein 88 (MyD88)[3]. The latter recruits and activates IL-1R-associated kinase 4 (IRAK4) that in turn associates with IRAK1 to promote its hyperphosphorylation[4, 5]. IRAK1 subsequently interacts with the E3 ubiquitin ligase TNF receptor-associated factor 6 (TRAF6) and TAK1 kinase that activates the IkappaB (IκB) kinase (IKK) complex to phosphorylate and promote proteasomal degradation of IκB proteins, key inhibitors of the NFκB transcription factor[6]. This allows for nuclear translocation of NFκB. TAK1 also triggers downstream activation of the p38, ERK, and JNK MAP kinase (MAPK) pathways that co-operate with NFκB to regulate the transcription of a plethora of pro-inflammatory genes including those encoding TNF, IL-1β, IL-18, and IL-6[7]. The inflammatory cytokines IL-1β and IL-18 are initially produced as inactive precursor proteins and require the generation of a signaling platform termed the inflammasome that processes IL-1β and IL-18 into their mature bioactive forms before being released from cells[8–10]. Thus, the secretion of mature IL-1β requires two signals. Firstly innate receptors, such as TLR4, is engaged by its ligand lipopolysaccharide (LPS) in Gram negative bacteria, to activate NFκB and induce transcription of the gene encoding the inactive pro-IL-1β precursor[11] with a second signal triggering inflammasome activation. Inflammasomes consist of a NLR protein such as NLRP3 that recruits the adapter protein ASC and caspase-1 into an oligomeric complex leading to auto-proteolytic processing of pro-caspase-1 into its active form that cleaves pro-IL-1β and pro-IL-18 precursors into their mature secreted forms[12, 13]. The activation of the inflammasome can also lead to a necrotic form of cell death termed pyroptosis[14]. The activators of NLRP3 include ATP, PAMPs, danger associated molecular patterns (DAMPs), and particulate substances. Additional NLR proteins, such as NLRC4, that is activated via NLR family apoptosis inhibitory proteins (NAIPs) and bacterial flagellin, can also form inflammasome complexes[15, 16]. Other inflammasome complexes include the cytosolic DNA sensor Absent in Melanoma-2 (AIM2) that associates with ASC and triggers inflammasome activation[17]. These canonical inflammasome pathways are also complemented by a non-canonical inflammasome in which murine caspase-11 (caspase-4 and caspase-5 in human cells) directly binds to cytosolic LPS[18] and is activated to induce pyroptosis in an analogous manner to caspase-1[19, 20]. While caspase-4, caspase-5, and caspase-11 cannot cleave pro-IL-1β or pro-IL-18, these caspases can promote the assembly of the NLRP3 inflammasome[19, 21, 22] to indirectly induce the production and secretion of the mature forms of IL-1β and IL-18.

There is much interest in delineating the molecular mechanisms that regulate inflammasome activation[23]. A number of findings have shown the significance of post-translational modification of components of the inflammasome with ubiquitination being of particular importance. TRIM31, FBXL2, and MARCH7 have been reported to promote ubiquitination and degradation of NLRP3[24–26] with NLRP3 needing to be de-ubiquitinated by the BRCC3 complex in order to facilitate NLRP3 activation[27–29]. However, there are currently no reports on E3 ubiquitin ligases directly contributing to activation of NLRP3. Here we show a critical role for the E3 ubiquitin ligase Pellino2 in regulating activation of the NLRP3 pathway.

Pellino proteins constitute a three-membered family of E3 ubiquitin ligases that have various regulatory roles in innate immune signaling pathways[3, 30, 31]. Originally identified as IRAK-interacting proteins, each family member shares an N-terminal forkhead-associated (FHA) domain that recognizes phospho-threonine residues and mediates association with IRAK1[32], and a C-terminal RING-like domain that confers E3 ubiquitin ligase activity and an ability to catalyze lysine 63 (K63)-linked poly-ubiquitination of substrate proteins such as IRAKs[33–35]. Pellino proteins are subject to phosphorylation by kinases, such as IRAK1 and TBK1/IKKε, that can enhance the E3 ligase activities of the Pellino proteins[35–39]. Murine genetic models have revealed roles for Pellino1 in TLR3 and TLR4 signaling pathways[40], as a suppressor of T cell activation[41, 42], a driver of neuroinflammation[43], and as a promoter of lymphomagenesis[44] and lung carcinogenesis[45]. We have previously generated Pellino3-deficient mice to identify important roles for Pellino3 as a negative regulator of TLR3 signaling[46] and TNF-induced cell killing[47], a mediator of NOD2 signaling in the gut[48] and a critical regulator of obesity-induced expression of IL-1β and insulin resistance[49].

While progress has been made in elucidating the functions of Pellino1 and Pellino3, there is a notable lack of insight into the physiological roles of Pellino2. Studies on Pellino2 have been largely confined to overexpression and knockdown approaches in cell lines that suggest a role for Pellino2 in IL-1/LPS-induced activation of ERK and JNK MAPK pathways and increased stabilization of mRNAs encoding pro-inflammatory proteins[50, 51]. We now generate Pellino2-deficient mice to reveal the first physiological role of Pellino2 as a mediator in the activation of the NLRP3 inflammasome pathway. We show that Pellino2 facilitates NLRP3-induced maturation and release of IL-1β and IL-18 in macrophages, while also promoting cell death by pyroptosis. The pathophysiological relevance of this role for Pellino2 is also shown by its contribution to the lethal effects of LPS in a murine model of endotoxemia and the in vivo induction of IL-1β in response to challenge with Gram negative bacteria. Pellino2 exerts its positive effects by promoting ubiquitination of NLRP3. We also identify for the first time that IRAK1 acts as an inhibitor of the NLRP3 pathway and reveal a counter-regulatory relationship between Pellino2 and IRAK1. In addition to revealing the first physiological role for Pellino2, this study also maps an entirely new regulatory pathway that controls NLRP3 activation.

## Results

**Pellino2 mediates mature IL-1β production in response to LPS.** In order to explore the physiological role for Pellino2 we generated Pellino2-deficient mice. Homologous recombination was used in embryonic stem cells to replace exons 2 to 6 in the *Peli2* gene with an F3-flanked selection cassette (Supplementary Fig. 1a). The selection cassette was subsequently removed by crossing mice with this targeted allele to mice expressing a *Flp* transgene thus generating mice with deletions of exons 2 to 6 of the *Peli2* gene, as confirmed by PCR analysis (Supplementary Fig. 1b). Semi-quantitative (Supplementary Fig. 1c) and real-time (Supplementary Fig. 1d) RT-PCR analysis of mRNA, isolated from bone marrow-derived macrophages (BMDMs), confirmed complete lack of expression of the *Peli2* gene in cells from *Peli2*$^{-/-}$ mice. Pellino2-deficient mice are viable and develop normally. Given the previously described roles for Pellino1 and Pellino3 in TLR signaling pathways, we initially compared the responsiveness of BMDMs from wild type (WT) and Pellino2-deficient mice to 24 h challenge with a wide array of TLR ligands, as measured by induction of the TLR-responsive proteins IL-6

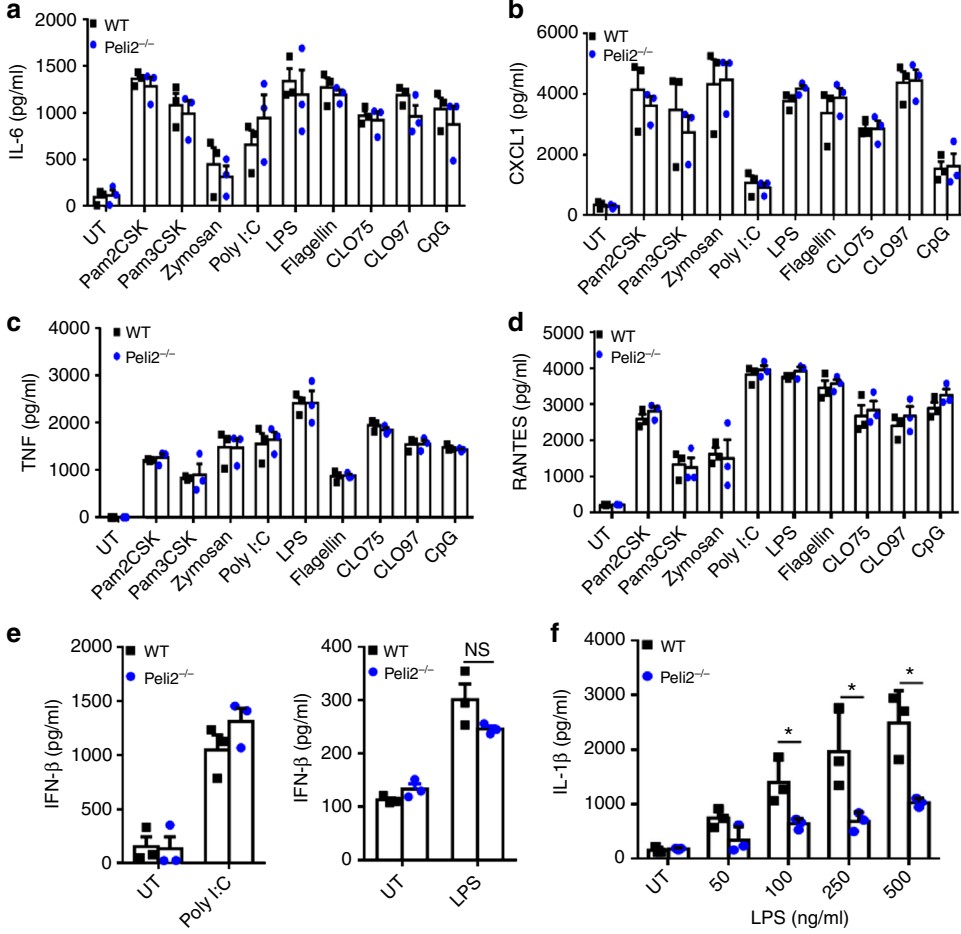

**Fig. 1** Pellino2 is required for production of IL-1β in response to LPS. **a–e** Enzyme-linked immunosorbent assay (ELISA) of **a** IL-6, **b** CXCL1, **c** TNF, **d** RANTES, and **e** IFN-β expression in medium from BMDMs isolated from wild-type (WT) and Pellino2-deficient (*Peli2*⁻/⁻) mice and treated with 10 ng/ml Pam2CSK, 10 ng/ml Pam3CSK, 1 μg/ml Zymosan, 25 μg/ml poly(I:C), 100 ng/ml lipopolysaccharide (LPS), 1 μg/ml flagellin, 1 μg/ml Clo75, 1 μg/ml Clo97, and 2 μg/ml CpG for 24 h. UT, untreated. **f** ELISA of IL-1β in medium from WT and Peli2⁻/⁻ BMDMs stimulated with the indicated concentrations of LPS for 24 h. *$p$ < 0.05 (paired, two-tailed Student's *t*-test). Data are presented as the mean of three independent experiments (**a-f**). Error bars, s.e.m.

(Fig. 1a), CXCL1 (Fig. 1b), TNF (Fig. 1c), RANTES (Fig.1d), and IFN-β (Fig. 1e). Each TLR ligand retained its ability to induce all of these cytokines in BMDMs lacking Pellino2 indicating its lack of physiological participation in these TLR signaling pathways. Since we have previously identified Pellino3 as an inhibitor of IL-1β expression, we next explored the role of Pellino2 in controlling the production of IL-1β. The levels of secreted IL-1β protein, in response to LPS, were strongly reduced in Pellino2-deficient BMDMs (Fig. 1f), suggesting a selective role for Pellino2 in the pathway controlling the production of IL-1β but not other pro-inflammatory proteins. However, Pellino2 is not involved in the transcriptional regulation of *Il1b* gene since LPS-induced expression of *Il1b* mRNA (Supplementary Fig. 2a) and pro-IL-1β protein (Supplementary Fig. 2b) is fully intact in Pellino2-deficient BMDMs. Indeed Pellino2 is dispensable for the early intracellular signaling pathways triggered by LPS that drive expression of the *Il1b* gene, since WT and *Peli2*⁻/⁻ BMDMs displayed the same patterns of time-dependent activation of NFκB and MAPK pathways as measured by phosphorylation of IκBα, p38 MAPK, JNK, and ERK (Supplementary Fig. 2c).

**Pellino2 mediates LPS-induced activation of NLRP3.** Given that Pellino2 is not required for induction of pro-IL-1β we next probed the role of Pellino2 in the activation of inflammasomes.

We initially employed the widely used two-signal model of LPS and ATP to evaluate the involvement of Pellino2 in the NLRP3 pathway. The priming of BMDMs from WT mice with LPS followed by stimulation with ATP resulted in strong expression and secretion of mature IL-1β, as measured by ELISA. However, this response was considerably reduced in cells from *Peli2*⁻/⁻ mice (Fig. 2a). Pellino2-deficient cells also exhibited reduced levels of IL-18 (Fig. 2b) and pyroptosis (Fig. 2c) relative to WT BMDMs, suggesting that Pellino2 plays a vital role in the NLRP3 pathway. This was further supported with immunoblotting analysis that demonstrated *Peli2*⁻/⁻ BMDMs to be less responsive than WT cells to LPS/ATP-induced processing of pro-IL-1β into its mature p17 IL-1β form and pro-caspase-1 into its active p20 form (Fig. 2d). β-Actin was used as a loading control in all immunoblotting analysis since its electrophoretic mobility is mid-range of the various proteins subjected to immunoblotting. Levels of processed IL-1β and caspase-1 were comparable between WT and *Peli2*⁻/⁻ BMDMs in response to double stranded DNA (poly (dA:dT), a trigger of the AIM2 inflammasome, suggesting some specificity for the NLRP3 pathway. To further confirm the role of Pellino2 in the NLRP3 pathway, another two-signal model, involving LPS as a prime stimulus followed by the potassium ionophore nigericin, was employed as an alternative approach to activate the NLRP3 pathway. Again *Peli2*⁻/⁻ BMDMs were much less responsive than WT cells with respect to LPS/nigericin-

induced production of mature IL-1β (Fig. 2e), IL-18 (Fig. 2f), pyroptosis (Fig. 2g), and processing of pro-IL-1β and pro-caspase-1 (Fig. 2h). While ATP and nigericin promotes NLRP3 activation by facilitating potassium efflux, particulate material like alum can also activate NLRP3. We next demonstrated that

Pellino2 is also required for alum-mediated secretion of IL-1β (Fig. 2i). These data are all consistent with a role for Pellino2 in activation of the NLRP3 pathway. In keeping with its potential targeting of NLRP3, the role of Pellino2 is independent on the priming stimulus. Using the TLR2 ligand zymosan, as an

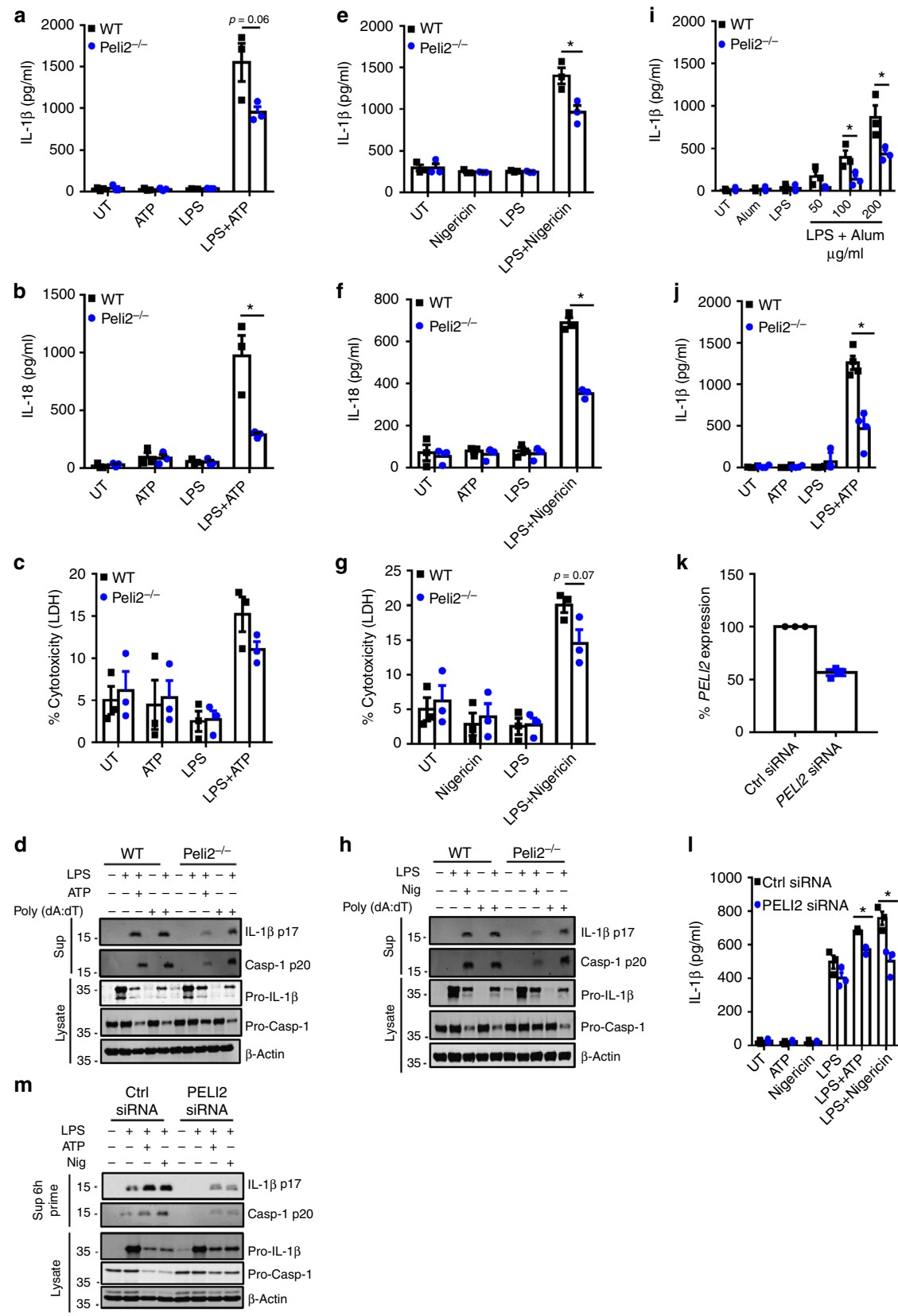

alternative first signal with ATP, resulted in comparable levels of zymosan-induced expression of *Il1b* mRNA (Supplementary Fig. 3a) and pro-IL-1β protein (Supplementary Fig. 3b) in WT and *Peli2*$^{-/-}$ cells. However, processing of pro-IL-1β (Supplementary Fig. 3c) and secretion of mature IL-1β (Supplementary Fig. 3d) in *Peli2*$^{-/-}$ BMDMs was severely impaired.

Having demonstrated a role for Pellino2 in the NLRP3 pathway in macrophages derived from bone marrow we wished to confirm a similar role in naturally occurring tissue resident macrophages. Peritoneal macrophages from *Peli2*$^{-/-}$ mice had less LPS/ATP induction of mature IL-1β than corresponding WT cells (Fig. 2j). Furthermore, we also confirm a role for Pellino2 in the NLRP3 pathway in human cells by showing that knockdown of *PELI2* expression in the human monocytic THP1 cell line (Fig. 2k) reduced LPS-, LPS/ATP-, and LPS/nigericin-induced secretion of mature IL-1β (Fig. 2l) and this is consistent with less processing of IL-1β and caspase-1 in *PELI2* knockdown cells (Fig. 2m).

We further examined the specificity of the role of Pellino2 for the NLRP3 inflammasome. Pellino2 is not employed in other canonical inflammasome pathways such as NLRC4 and AIM2 since respective triggers of these inflammasomes, flagellin and poly (dA:dT), had comparable efficacy in inducing IL-1β in WT and *Peli2*$^{-/-}$ BMDMs (Supplementary Fig. 4a). Furthermore, Pellino2 was dispensable for non-canonical inflammasome activation as Pellino2 deficiency did not affect the ability of cytosolic LPS to trigger processing of mature IL-1β (Supplementary Fig. 4b) or pyroptosis (Supplementary Fig. 4c). Additionally, cholera toxin B (CTB) co-administration with LPS provides another way to activate the non-canonical inflammasome[19]. Again LPS/CTB-induced release of mature IL-1β (Supplementary Fig. 4d) and pyroptosis (Supplementary Fig. 4e) were unaltered in BMDMs lacking Pellino2. Together, these data suggest that Pellino2 is not involved in non-canonical inflammasome activation and instead serves a selective role in the canonical NLRP3 pathway.

**Pellino2 contributes to LPS-induced lethality.** We next aimed to explore the physiological relevance of the role for Pellino2 in responding to LPS by extending our studies into in vivo models. Consequently, WT and *Peli2*$^{-/-}$ mice were subjected to intraperitoneal administration of non-lethal dose of LPS and serum levels of a range of cytokines were assayed. LPS-induced increased serum levels of IL-6 (Fig. 3a), TNF (Fig. 3b), CXCL1 (Fig. 3c), RANTES (Fig. 3d), and IL-1β (Fig. 3e) in WT mice and whereas the same levels of IL-6, TNF, CXCL1, and RANTES were observed in Pellino2-deficient mice, the serum levels of IL-1β in *Peli2*$^{-/-}$ mice were reduced relative to WT mice (Fig. 3e). These data are consistent with our earlier findings above that indicated a role for Pellino2 in controlling IL-1β maturation. The

pathophysiological relevance of this role was further evaluated by administering mice with lethal dose of LPS and survival monitored over a 72 h period post administration. All WT mice succumbed to the lethal effects of LPS after 48 h but *Peli2*$^{-/-}$ mice had a 50% survival rate up to 72 h (Fig. 3f) suggesting that Pellino2 plays an important in vivo role in mediating IL-1β production in response to LPS and in promoting LPS-induced lethality in endotoxaemia.

**Pellino2 mediates NLRP3 anti-bacterial responses.** We extended our studies on the physiological relevance of the role of Pellino2 in responding to LPS and in promoting inflammasome activation by comparing the responsiveness of BMDMs from WT and *Peli2*$^{-/-}$ mice to the Gram negative bacteria *Citrobacter rodentium* (*C. rodentium*) and *Escherichia coli* (*E. coli*). Although these bacteria activate the NLRP3 inflammasome they can also induce caspase-11 expression and synergize with activated NLRP3 to process IL-1 family cytokines and instigate pyroptosis[52, 53]. *C. rodentium* induced production of mature IL-1β (Fig. 4a) and caspase-11 (Fig. 4b) in WT BMDMs and whereas the levels of IL-1β were reduced in *Peli2*$^{-/-}$ cells, the expression of caspase-11 was intact. Similarly, *E.coli* induced IL-1β maturation in a Pellino2-dependent manner (Fig. 4c) whereas caspase-11 expression was independent of Pellino2 (Fig. 4d) suggesting a direct and specific role for Pellino2 in the NLRP3 pathway. Pellino2 also mediated the production of IL-1β and IL-18 but not CXCL1 (Fig. 4e) and processing of pro-caspase-1 (Fig. 4f) in response to another Gram negative bacteria *Pseudomonas aeruginosa* (*P. aeruginosa*). This again supports a role for Pellino2 in bacterial-induced inflammasome activation. The physiological relevance of this role was confirmed by demonstrating that *P. aeruginosa*-infected *Peli2*$^{-/-}$ mice produced less IL-1β and IL-18 but the same levels of another pro-inflammatory cytokine IL-6 compared with infected WT mice (Fig. 4g).

**Pellino2 FHA and RING domains facilitate NLRP3 activation.** Having provided strong supporting evidence for a role for Pellino2 in the NLRP3 inflammasome, we next probed the underlying mechanism. Our initial mechanistic studies focused on Pellino2 itself and the functional relevance of its FHA and RING-like domains for NLRP3 activation. To this end, we used murine stem cell virus (MSCV) transduction to re-constitute *Peli2*$^{-/-}$ BMDMs with WT murine Pellino2 and mutated forms with point mutations in its FHA or RING-like domain (Fig. 5a). The re-introduction of WT murine Pellino2 into *Peli2*$^{-/-}$ BMDMs reconstituted the ability of LPS/ATP to induce IL-1β and IL-18 to levels that were equivalent to those observed in WT BMDMs (Fig. 5b). However, the FHA mutant form of Pellino2, that lacks substrate binding activity, and the RING mutant form, that lacks E3 ubiquitin ligase activity, failed to re-constitute LPS/ATP

**Fig. 2** Pellino2 mediates activation of the NLRP3 pathway. ELISA of **a** IL-1β and **b** IL-18, and **c** LDH assay of medium from WT and *Peli2*$^{-/-}$ BMDMs treated with 100 ng/ml LPS for 3 h with or without further stimulation with 2.5 mM ATP for 1 h. UT, untreated. **d** Immunoblot analysis of IL-1β and Caspase-1 in medium (Sup) and lysates from WT and *Peli2*$^{-/-}$ BMDMs stimulated with 100 ng/ml LPS for 3 h with or without further stimulation with 2.5 mM ATP for 1 h or transfection of Poly (dA:dT) (1 μg/ml) for 6 h. ELISA of **e** IL-1β and **f** IL-18 and **g** LDH assay of medium from WT and Peli2$^{-/-}$ BMDMs treated with 100 ng/ml LPS for 3 h with or without further stimulation with 5 mM Nigericin for 1 h. **h** Immunoblot analysis of IL-1β and Caspase-1 in medium (Sup) and lysates from WT and *Peli2*$^{-/-}$ BMDMs stimulated with 100 ng/ml LPS for 3 h with or without further stimulation with 5 mM Nigericin for 1 h or transfection of Poly (dA:dT) (1 μg/ml) for 6 h. **i** ELISA of IL-1β of medium from WT and *Peli2*$^{-/-}$ BMDMs treated with 100 ng/ml LPS for 3 h with or without further stimulation with the indicated concentrations of Alum for 6 h. **j** ELISA of IL-1β in medium from peritoneal-resident macrophages isolated from WT and Peli2$^{-/-}$ mice. Cells were treated with 100 ng/ml of LPS for 3 h with or without further stimulation with 2.5 mM ATP for 1 h. **k-m** Human THP1 cells were transfected with human Pellino2-specific siRNA or control siRNA. **k** Quantitative RT-PCR analysis of PELI2 expression in transfected cells. **l** ELISA of IL-1β in medium from transfected THP1 cells stimulated with 100 ng/ml LPS for 6 h with or without further treatment with 2.5 mM ATP or 5 mM Nigericin for 1 h. **m** Immunoblot analysis of IL-1β and Caspase-1 in medium (Sup) and lysates from transfected THP1 cells stimulated with 100 ng/ml LPS for 6 h with or without further stimulation with 2.5 mM ATP or 5 mM Nigericin for 1 h. β-Actin was used as loading controls. *$p < 0.05$ (paired, two-tailed Student's *t*-test). Data are biological replicates that are representative of three independent experiments. Error bars, s.e.m.

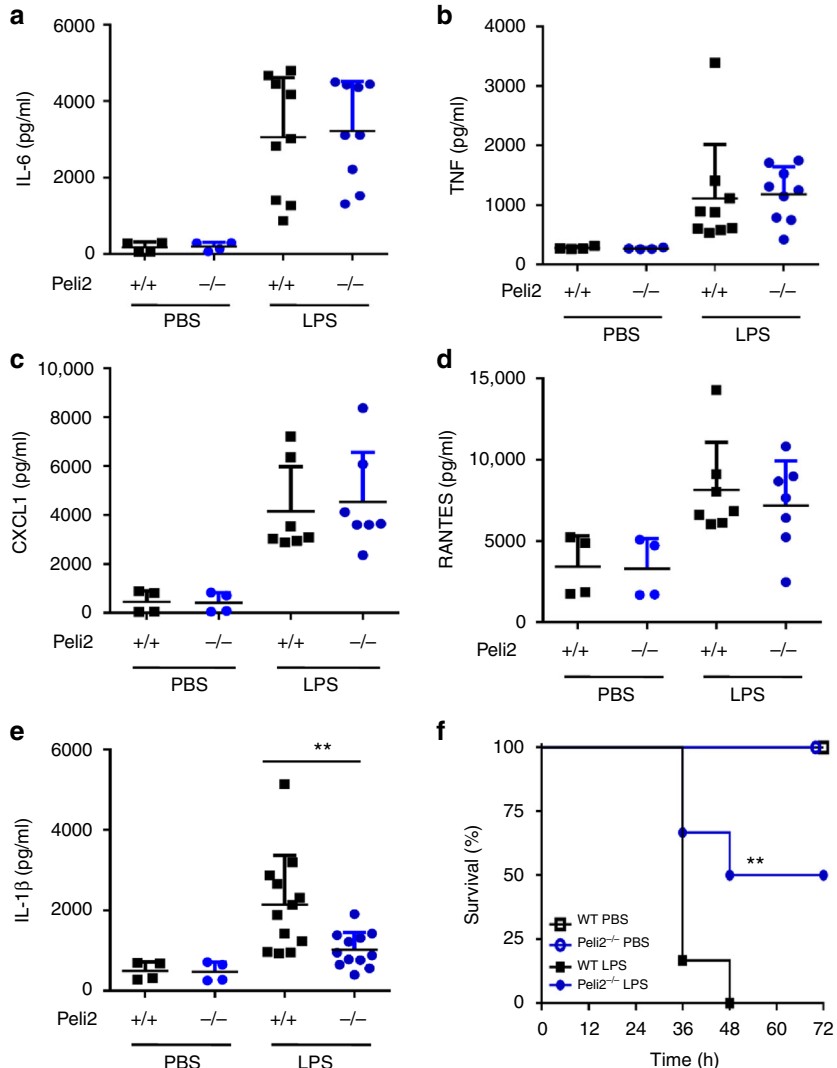

**Fig. 3** Pellino2 is required for in vivo expression of IL-1β and lethality in response to LPS. ELISA of **a** IL-6, **b** TNF, **c** CXCL1, **d** RANTES, and **e** IL-1β, from serum of WT and *Peli2*[−/−] mice at 16 h post-intraperitoneal injection with 20 mg/kg LPS or PBS (*n* = 7–12). **f** Survival rates of age- and sex-matched WT (*n* = 12) and *Peli2*[−/−] (*n* = 12) mice after intraperitoneal injection with 50 mg/kg of LPS or PBS (*n* = 4 for each of WT and *Peli2*[−/−]) as a vehicle control. Mice were monitored every 6 h for 72 h. **P < 0.01 (paired, two-tailed Student's *t*-test (**e**) and log-rank Mantel-Cox test (**f**)). Data indicate samples from individual mice (*n* = 4 for all PBS groups (**a–f**), *n* = 9 for LPS groups (**a**), *n* = 8 for LPS/WT, *n* = 9 for LPS/ *Peli2*[−/−] (**b**), *n* = 7 for LPS groups (**c, d**), *n* = 11 for LPS/ WT, *n* = 12 for LPS/*Peli2*[−/−] (**e**). Data are pooled biological replicates from two independent experiments. Error bars, s.e.m.

function in *Peli2*[−/−] BMDMs. These data indicate that Pellino2 is dependent on both its FHA and RING-like domains in order to trigger activation of the NLRP3 pathway. This is further supported by the observation that the re-introduction of WT Pellino2 could re-constitute the LPS/ATP-induced processing of pro-caspase-1 in *Peli2*[−/−] BMDMs to the same degree as displayed in WT cells whereas both the FHA and RING-like mutants of Pellino2 were ineffective in this regard (Figs. 5c, d).

**Pellino2 is required for NLRP3-induced ASC oligomerization.** We next aimed to explore how Pellino2 can integrate into the NLRP3 pathway. The production of mROS has been proposed as a general converging mechanism for various NLRP3 stimuli. However, WT and *Peli2*[−/−] BMDMs produced the same levels of mitochondrial ROS under conditions of NLRP3 activation with LPS and ATP (Supplementary Fig. 5). This suggested that Pellino2 acts independently of such a general triggering mechanism and so we focused on more specific protein targets in the NLRP3 pathway. Given the crucial role of ASC in the NLRP3

inflammasome and since the assembly of the NLRP3 inflammasome is associated with oligomerization of ASC, that manifests as a single ASC speck in cells, we next characterized the effects of Pellino2 deficiency on ASC aggregation under conditions of NLRP3 activation. ASC speck formation was detectable in WT BMDMs that were primed with LPS and then stimulated with ATP but the number of cells displaying ASC formation under these conditions was significantly reduced in *Peli2*[−/−] BMDMs (Fig. 6a). Since ASC speck formation reflects oligomerization of ASC, we treated cell lysates with a bifunctional chemical cross-linking agent and subjected samples to ASC immunoblotting to assess for oligomerization. LPS/ATP-induced strong oligomerization of ASC in WT BMDMs but the levels of higher molecular weight ASC oligomers were reduced in *Peli2*[−/−] BMDMs (Fig. 6b), in keeping with impaired ASC speck formation under the same conditions. To further confirm the role of Pellino2 in mediating NLRP3-induced ASC aggregation, we used LPS/ nigericin as a second means to activate the NLRP3 inflammasome. LPS priming, followed by nigericin stimulation, promoted ASC oligomerization in WT BMDMs but this was again reduced

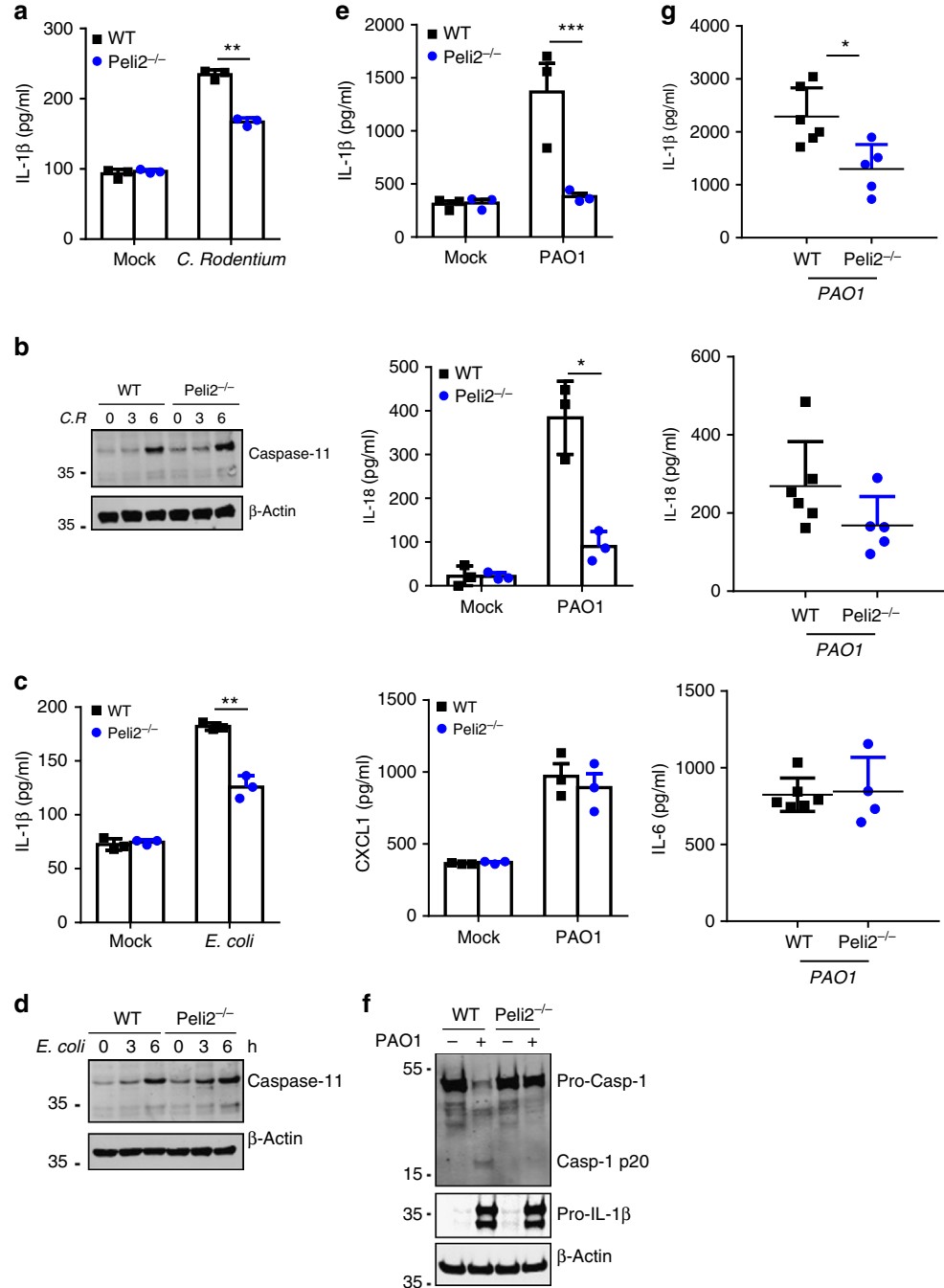

**Fig. 4** Pellino2 mediates activation of the NLRP3 pathway in response to bacterial infection. **a-f** WT and $Peli2^{-/-}$ BMDMs were infected with **a, b** C. rodentium, **c, d** E.coli, or **e, f** P. aeruginosa (PA01 strain) at a multiplicity of infection (MOI) of 100. **a, c, e** ELISA of IL-1β, IL-18, and CXCL1 in medium from BMDMs infected for 6 h. **b, d** Immunoblot analysis of Caspase-11 in lysates from cells infected for 0-6 h. **f** Immunoblot analysis of IL-1β and Caspase-1 in lysates from cells infected with PAO1 for 3 h. β-Actin was used as loading controls. **g** ELISA of IL-1β, IL-18, and IL-6 in peritoneal lavage from WT and $Peli2^{-/-}$ mice previously infected for 10 h by intraperitoneal injection of PAO1 ($1.5 \times 10^{7}$ CFU). *$p < 0.05$, **$p < 0.01$ (paired, two-tailed Student's $t$-test). Data represent the mean ± s.e.m. of three independent experiments (**a, c, e**), biological replicates that are representative of 3 independent experiments (**b, d, f**), or samples from individual mice ($n = 4$–6 per group) (**g**)

in $Peli2^{-/-}$ BMDMs (Fig. 6c) consistent with Pellino2 facilitating ASC oligomerization in response to NLRP3 activation. The impaired NLRP3-induced oligomerization of ASC in $Peli2^{-/-}$ BMDMs was overcome by re-constituting these cells with WT Pellino2 (Fig. 6d). However, mutated forms of Pellino2 with point mutations in its FHA or RING-like domains failed to re-constitute this function of Pellino2. This correlates closely with the earlier re-constitution of NLRP3-induced IL-1β and IL-18

expression with WT Pellino2 but not mutated forms (Fig. 5b) strongly indicating that Pellino2-mediated ASC oligomerization translates into downstream production of mature bioactive forms of IL-1β and IL-18.

**Pellino2 promotes ubiquitination of NLRP3.** Given that Pellino2 appears to act upstream of ASC, we next investigated

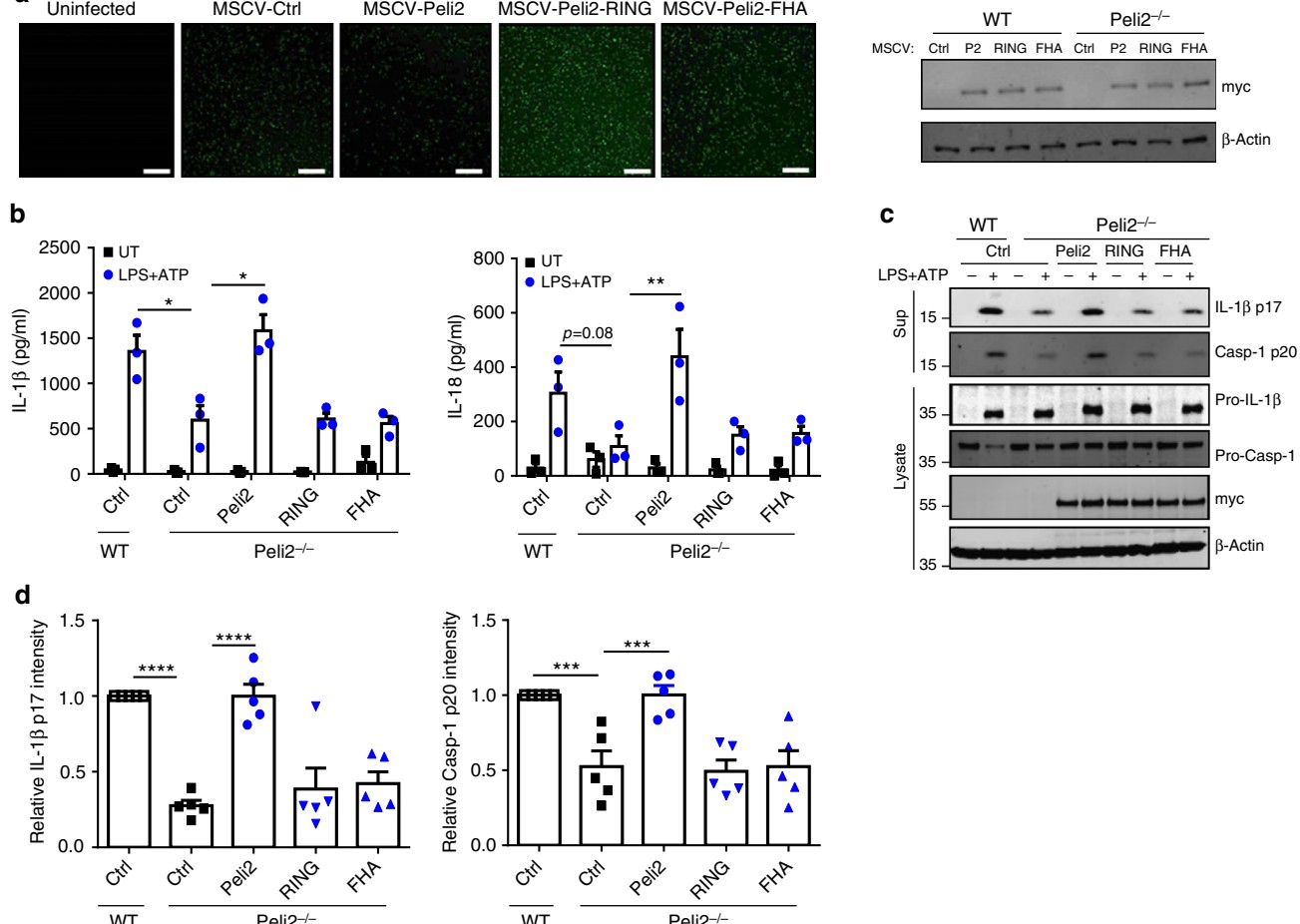

**Fig. 5** The activation of the NLRP3 pathway is dependent on the FHA and RING-like domain of Pellino2. WT and *Peli2*$^{-/-}$ BMDMs were infected with GFP-expressing murine stem cell virus (MSCV) as control (Ctrl) or with MSCV containing an expression construct encoding myc-tagged murine Pellino2 (Peli2), Pellino2 RING mutant (Peli2-RING), or Pellino2 FHA mutant (Peli2-FHA). **a** GFP expression (left panels) and immunoblot analysis of myc (right panel) (scale bar = 20 μm). **b** ELISA of IL-1β and IL-18 in medium from virus-infected WT and Peli2$^{-/-}$ BMDMs stimulated with 100 ng/ml LPS for 3 h followed by 2.5 mM ATP for 1 h. **c, d** Immunoblot analysis (**c**) and densitometry analysis (**d**) of IL-1β and Caspase-1 in medium (Sup) and cell lysates from virus-infected WT and Peli2$^{-/-}$ BMDMs stimulated with 100 ng/ml LPS for 3 h followed by 2.5 mM ATP for 1 h. *$P < 0.05$; **$P < 0.01$; ***$P < 0.001$; ****$P < 0.0001$ (paired, two-tailed Student's *t*-test (**b**) or two-way ANOVA (**d**)). Data are biological replicates that are representative of 3–5 independent experiments (**a, c**) or represent the mean ± s.e.m. of 3–5 independent experiments (**b, d**)

NLRP3 as a potential direct target for Pellino2. We were especially interested in characterizing the regulatory mechanisms controlling ubiquitination of NLRP3 since earlier studies had shown NLRP3 to be subject to ubiquitination and our data above indicated the mediatory role of Pellino2 to be dependent on the RING-like domain that underpins its E3 ubiquitin ligase activity. We thus treated WT and *Peli2*$^{-/-}$ BMDMs with LPS for various times, immunoprecipitated NLRP3 and analyzed the ubiquitination status by Western blotting. LPS promoted time-dependent polyubiquitination of NLRP3 in WT cells and this was considerably reduced in *Peli2*$^{-/-}$ BMDMs (Fig. 7a) indicating that LPS induces ubiquitination of NLRP3 in a Pellino2-dependent manner. We next defined the type of linkages in the polyubiquitin chains that are attached to NLRP3. Thus, using K63-specific TUBEs we isolated K63-linked ubiquitinated proteins followed by immunoblotting for NLRP3 to show that LPS promotes time dependent K63-linked ubiquitination of NLRP3 in WT macrophages and this is reduced in Pellino2-deficient cells (Fig. 7b). Similar approaches using K48-specific TUBEs failed to detect any K48-linked ubiquitination of NLRP3 in response to LPS (Fig. 7c) demonstrating that LPS promotes K63-linked ubiquitination of NLRP3 in a Pellino2-dependent manner. While previous reports

have described a requirement for NLRP3 to be de-ubiquitinated in order to trigger downstream effects of inflammasome activation, our data lead us to hypothesize that part of the priming for NLRP3 inflammasome activation by LPS may be related to a requirement to initially induce ubiquitination of NLRP3. We present further evidence in support of this hypothesis by demonstrating that MCC950, a highly selective inhibitor of NLRP3[54], strongly suppresses LPS/ATP induction of IL-1β in WT BMDMs (Fig. 7d) and also inhibits LPS-induced ubiquitination of NLRP3 (Fig. 7e). Interestingly, while MCC950 strongly suppressed the ability of LPS to promote ubiquitination of NLRP3, MCC950 also appeared to cause some basal ubiquitination of NLRP3 in unstimulated cells. In order to address these apparent contrasting roles of MCC950 we characterized the effects of MCC950 on formation of polyubiquitin chains of different linkages. Using K63-specific TUBEs to isolate K63-linked ubiquitinated proteins we revealed LPS-induced K63-linked ubiquitination of NLRP3 and this was fully abrogated in cells pretreated with MCC950 (Fig. 7f). Notably, MCC950 alone failed to induce any K63-linked ubiquitination of NLRP3 suggesting that the basal ubiquitination of NLRP3 observed in response to MCC950 (Fig. 7e) is due to ubiquitin chains that are joined by

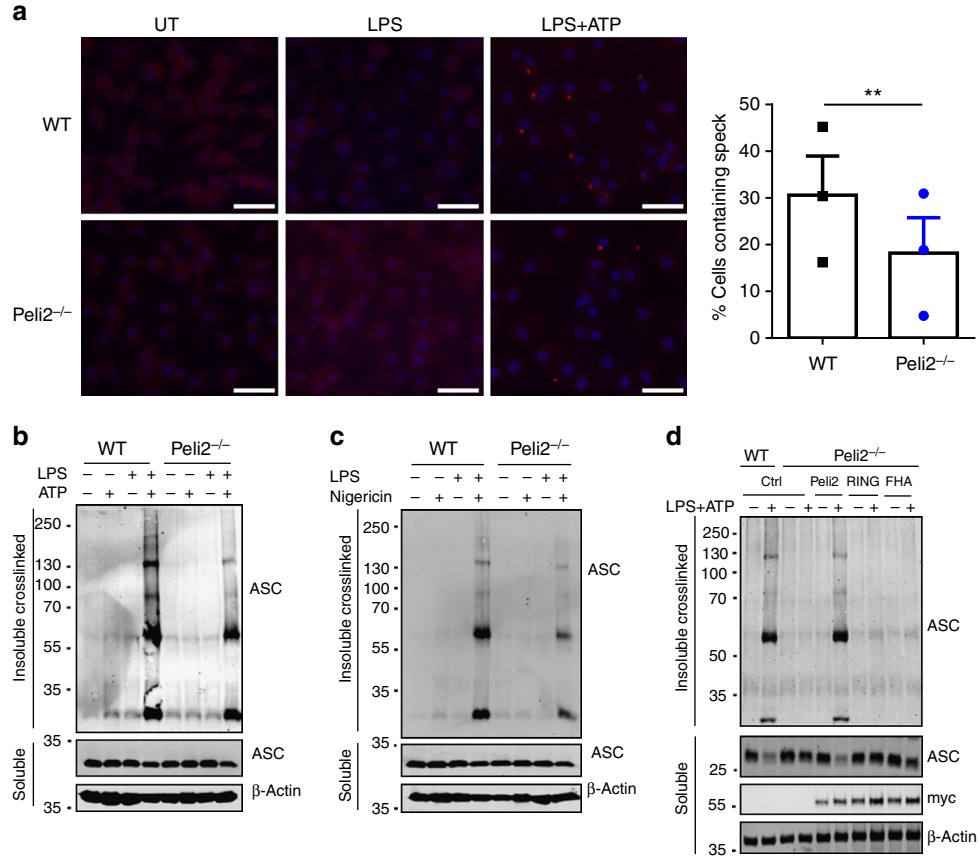

**Fig. 6** Pellino2 mediates NLRP3-dependent oligomerization of ASC. **a** Immunofluorescence staining of ASC in WT and *Peli2*−/− BMDMs that were left untreated (UT) or  treated with 100 ng/ml LPS for 3 h and further stimulated with ATP for 30 min. ASC specks were detected by immunostaining using anti-ASC antibody and anti-rabbit Alexa Fluor 568 (red) and cells were counter stained with nuclei-staining DAPI. The histogram quantitates the percentage of cells that exhibit ASC speck formation. (scale bar = 100 μm). **b, c** Immunoblot analysis of ASC in chemically cross-linked NP-40 insoluble fractions and in NP-40 soluble fractions from cell lysates of WT and Peli2−/− BMDMs stimulated with 100 ng/ml LPS for 3 h with or without further treatment with **b** 2.5 mM ATP, or **c** 5 mM Nigericin for 30 min. β-Actin was used as loading controls. **d** WT and *Peli2*−/− BMDMs were infected with MSCV as control (Ctrl) or with MSCV containing an expression construct encoding myc-tagged murine Pellino2 (Peli2), Pellino2 RING mutant (RING), or Pellino2 FHA mutant (FHA). Immunoblot analysis of ASC in chemically cross-linked NP-40 insoluble fractions and in NP-40 soluble fractions from cell lysates of MSCV-infected cells treated with 100 ng/ml LPS for 3 h followed by 2.5 mM ATP for 30 min. The expression of the Pellino2 constructs was measured by immunoblotting with an anti-myc antibody. **\*\***$p < 0.01$ (paired, two-tailed Student's *t*-test). Data are the mean ± s.e.m. of three independent experiments (**a**, right panel) or biological replicates that are representative of three independent experiments (**a** left panel, **b, c, d**)

different linkages. Thus, LPS promotes transient K63-linked ubiquitination of NLRP3, that may prime NLRP3 for subsequent activation and this is countered by the inhibitory effects of MCC950 that oppose the K63-linked ubiquitination of NLRP3. These data arising from the use of MCC950 are consistent with our hypothesis of a requirement for LPS to prime K63-linked ubiquitination of NLRP3 for its later activation and suggests that impaired ubiquitination of NLRP3 in response to LPS in *Peli2*−/− BMDMs may underlie the reduced activation of NLRP3 in these cells. Indeed MCC950 does not affect the residual LPS/ATP-induced levels of IL-1β in *Peli2*−/− BMDMs (Fig. 7d) raising the possibility that the targeting of NLRP3 ubiquitination may contribute to the inhibitory effects of MCC950.

We next characterized the nature of the functional relationship between Pellino2 and NLRP3. Using *Peli2*−/− BMDMs, reconstituted with a myc-tagged form of Pellino2 that we described above to restore NLRP3 activation, we demonstrated that LPS induces a time-dependent interaction of Pellino2 with NLRP3 (Fig. 7g). We assessed the potential of Pellino2 to act as a direct E3 ligase for NLRP3. However, the co-expression of Pellino2 and NLRP3 with WT ubiquitin (Supplementary Fig. 6a) or a mutant ubiquitin form containing a single lysine at residue 63 (K63A)

(Supplementary Fig. 6b) failed to show any Pellino2-induced ubiquitination of NLRP3. This suggested that Pellino2 may be exerting its effects on NLRP3 in a more indirect manner and possibly, by targeting a protein that directly regulates NLRP3.

**IRAK1 inhibits NLRP3 and is targeted by Pellino2**. Given that we have previously shown Pellino2 to interact with IRAK1 and the latter can associate with NLRP3 we investigated the ability of Pellino2 to target IRAK1 and regulate its binding to NLRP3. In keeping with previous studies we show that Pellino2 can interact with IRAK1 (Fig. 8a) and promote ubiquitination of IRAK1 in a RING domain-dependent manner (Fig. 8b) when these proteins are co-expressed in HEK293T cells. LPS also induced time-dependent ubiquitination of IRAK1 in WT BMDMs and this was suppressed in *Peli2*−/− cells (Fig. 8c) highlighting Pellino2 as an important mediator of IRAK1 ubiquitination in response to LPS. While such ubiquitination by LPS in WT cells is followed by degradation of most of the pool of IRAK1, the greatly reduced ubiquitination of IRAK1 in *Peli2*−/− cells still permitted degradation of IRAK1, albeit with slightly delayed kinetics and more residual IRAK1 than in WT cells. Furthermore, using TUBEs to

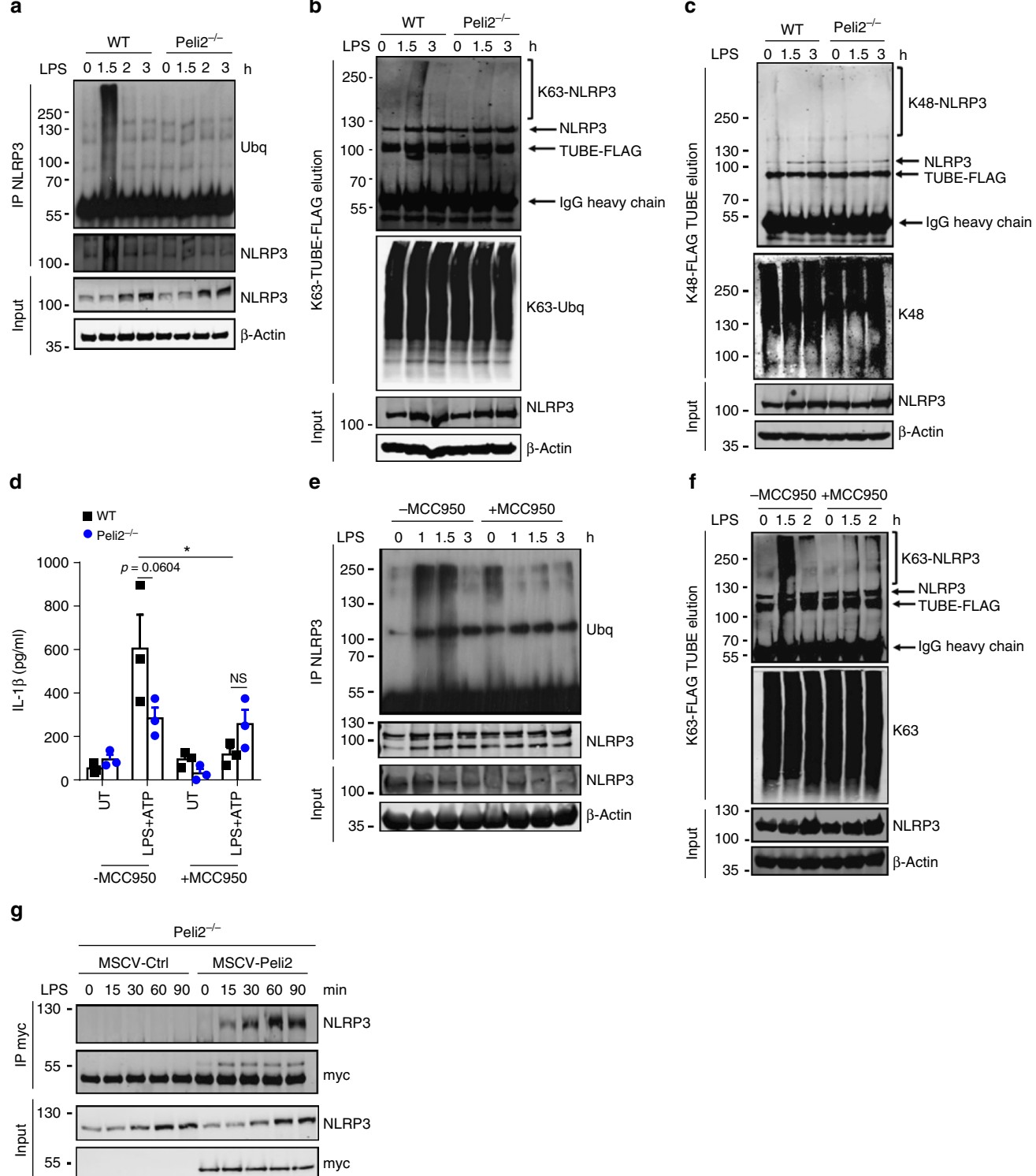

**Fig. 7** Pellino2 mediates LPS-induced ubiquitination and activation of NLRP3. **a–c** Immunoblot analysis of NLRP3 and ubiquitin in cell lysates (Input) and immunoprecipitated (IP) NLRP3 samples (**a**) or NLRP3 and K63-linked ubiquitin (K63-ubq) in K63-TUBE-FLAG elution and cell lysates (**b**) or NLRP3 and K48-ubq in K48-TUBE-FLAG elution and cell lysates (**c**) from WT and *Peli2⁻/⁻* BMDMs treated with **a** 100 ng/ml LPS for the indicated times. **d** ELISA of IL-1β in medium from WT and Peli2⁻/⁻ BMDMs treated with 100 ng/ml LPS for 2 h followed by sequential treatment with 1 µM MCC950 for 1 h and 2.5 mM ATP for 30 min. UT, untreated. **e, f** Immunoblot analysis of ubiquitin and NLRP3 in lysates (Input) and immunoprecipitated (IP) NLRP3 samples (**e**) or NLRP3 and K63-ubq in K63-TUBE-FLAG elution and cell lysates (**f**) from WT and *Peli2⁻/⁻* BMDMs pre-treated with 1 µM MCC950 for 1 h followed by treatment with 100 ng/ml LPS for the indicated times. **g** *Peli2⁻/⁻* BMDMs were infected with MSCV as control (MSCV-Ctrl) or with MSCV containing an expression construct encoding myc-tagged murine Pellino2 (MSCV-Peli2). Immunoblot analysis of NLRP3 and myc in lysates (Input) and immunoprecipitated (IP) myc samples from MSCV-infected cells treated with 100 ng/ml LPS for the indicated times. β-Actin was used as loading controls. *$p < 0.05$ (paired, two-tailed Student's *t*-test). NS, not significant. Data are biological replicates that are representative of three independent experiments (**a–c, e–g**) or mean ± s.e.m. of three independent experiments (**d**)

isolate ubiquitinated proteins we again demonstrated that LPS-induced ubiquitination of IRAK1 was reduced in *Peli2*−/− cells (Fig. 8d). We next assessed the effects of the loss of IRAK1 ubiquitination in *Peli2*−/− cells on the ability of IRAK1 to interact with NLRP3. LPS promoted time-dependent interaction of IRAK1 with NLRP3 in WT BMDMs but this interaction was augmented in *Peli2*−/− cells (Fig. 8e) suggesting that Pellino2-mediated ubiquitination of IRAK1 may impair the binding of the latter to NLRP3. We were thus keen to study the functional relevance of IRAK1 binding to NLRP3 and compared NLRP3 activation between WT and IRAK1-deficient BMDMs. Intriguingly the absence of IRAK1 resulted in further enhancement of the levels of mature IL-1β produced in response to LPS/ATP or LPS/nigericin

(Fig. 8f, left panel), two regimes to trigger the NLRP3 inflammasome. In contrast, the absence of IRAK1 leads to reduced levels of TNF (Fig. 8f, right panel) in keeping with the early receptor proximal role of IRAK1 in the TLR4 pathway that drives induction of TNF[55]. This suggests that IRAK1 acts to negatively regulate NLRP3 and this was further supported by augmented processing of pro-IL-1β and pro-caspase-1 in IRAK1-deficient BMDMs (Fig. 8g). Similar findings were also observed in IRAK1-deficient immortalized BMDMs (Supplementary Fig. 7a, b) with these studies also including control experiments with IRAK4-deficient cells that produced barely detectable levels of secreted IL-1β (Supplementary Fig. 7a). The latter is likely due to IRAK4 being an essential mediator of the LPS priming of pro-IL-1β production.

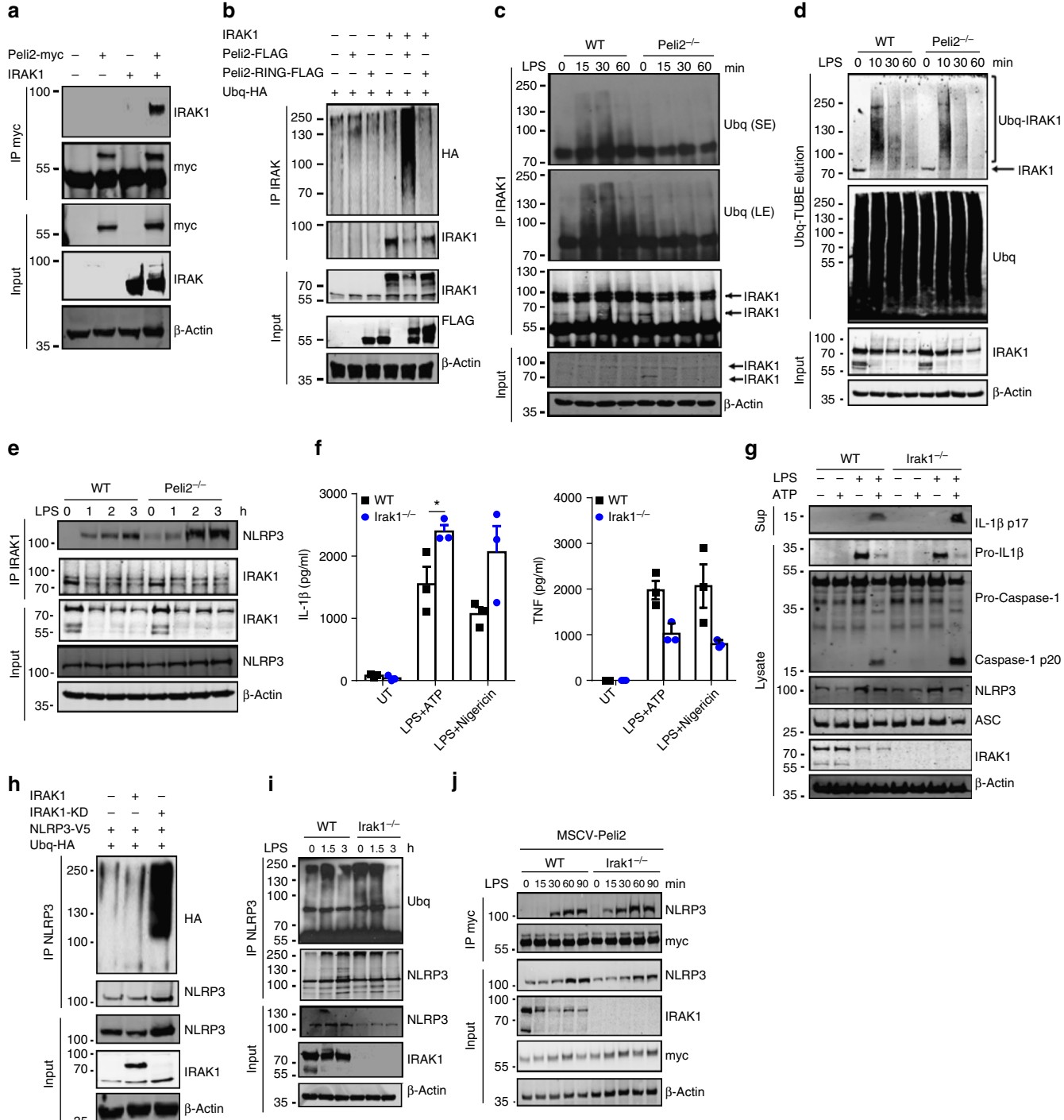

We next investigated if IRAK1 could regulate the ubiquitination of NLRP3 since IRAK1 is dispensable for signal 1-mediated induction of pro-IL-1β. We initially co-expressed IRAK1 and NLRP3 and while it proved difficult to detect any ubiquitination of NLRP3 when expressed alone or with IRAK1, intense ubiquitination of NLRP3 was observed when co-expressed with a kinase dead form of IRAK1 (Fig. 8h). Since the latter likely acts as a dominant negative inhibitor of endogenous IRAK1, the study suggested that the kinase activity of IRAK1 may act to suppress NLRP3 ubiquitination. We also compared LPS-induced ubiquitination of NLRP3 in WT and IRAK1-deficient BMDMs and observed enhanced NLRP3 ubiquitination in the absence of IRAK1 (Fig. 8i). Intriguingly the enhanced ubiquitination of NLRP3 in IRAK1-deficient cells is also associated with increased binding of Pellino2 to NLRP3 (Fig. 8j) again supporting a role for Pellino2-dependent ubiquitination of NLRP3 being an important part of LPS priming of the NLRP3 inflammasome. These data are consistent with a model of NLRP3 activation that is counter-regulated by IRAK1 and Pellino2 (Fig. 9). Thus, we propose that IRAK1 binds to NLRP3 to suppress ubiquitination and priming of NLRP3. Pellino2 can counter this braking mechanism with these effects of Pellino2 being associated with facilitation of IRAK1 ubiquitination, dissociation of IRAK1 from NLRP3 and ubiquitination of NLRP3 in a Pellino2-dependent manner. While over-expression studies have previously shown Pellino2 to promote ubiquitination of IRAK1[35], in vitro ubiquitination assays with purified forms of Pellino2 and IRAK would be required to confirm that IRAK1 is a direct substrate for Pellino2. Overall, our data demonstrate a role for Pellino2 in the priming phase of the NLRP3 pathway that facilitates NLRP3 inflammasome assembly, ASC oligomerization and downstream caspase-1-mediated processing of pro-IL-1β and pro-IL-18.

## Discussion

This study identifies the first physiological role of Pellino2 and a new regulatory network that controls the NLRP3 inflammasome. The findings highlight an important role for Pellino2 in NLRP3 activation by promoting the K63-linked ubiquitination of NLRP3 as part of the priming phase. We also propose that IRAK1 can negatively regulate ubiquitination of NLRP3 and its downstream signaling and Pellino2 can counter this effect by ubiquitinating IRAK1 thus impairing interaction of the latter with NLRP3 and relieving the braking system of IRAK1 on the NLRP3 inflammasome. The need for such complex regulatory control of the NLRP3 inflammasome is likely due to the damaging consequences of unregulated NLRP3 activity in response to infection and also in auto-inflammatory diseases[56].

There is an emerging appreciation of the physiological roles of Pellino proteins. Pellino1 plays important roles in TLR3 and TLR4 pathways[40], T cell activation[41], CNS inflammation[43], and cancer generation[44, 45]. We have previously shown Pellino3 to negatively regulate TLR3 signaling[46], suppress TNF-induced cell death[47], regulate IL-1β expression to control obesity-induced insulin resistance[49], and mediate NOD2 signaling to facilitate gut homeostasis[48]. However, until this present study, there have been no reports on the physiological role of Pellino2. We now propose that it plays an important mediatory role in assembly and activation of the NLRP3 inflammasome. Pellino2 is not involved in TLR-induced activation of NFκB or induction of pro-IL-1β or NLRP3 but instead facilitates the K63-linked ubiquitination of NLRP3 and we propose this as a novel pathway in the priming process. The lack of role of Pellino2 in the NFκB pathway and other early signaling pathways such as MAPK cascades is consistent with its absence of involvement in TLR-induced expression of pro-inflammatory proteins in macrophages. Interestingly previous studies using overexpression and gene knockdown approaches in cell lines had proposed a role for Pellino2 in mediating LPS-induced activation of MAPK pathways to regulate pro-inflammatory gene expression[50, 51]. However, we show that these pathways are intact and functional in Pellino2-deficient cells suggesting that its role, if any, is dispensable in these signal transduction pathways. This may be due to functional redundancy with other E3 ubiquitin ligases and indeed a recent report suggests some redundancy between Pellino1, Pellino2, and TRAF6, at least with respect to IL-1 signaling[57]. However, the impaired activation of NLRP3 activation in Pellino2-deficient cells clearly indicates a critical role for Pellino2 in inflammasome biology that cannot be fully served by other E3 ubiquitin ligases. Interestingly, Pellino2 joins Pellino3 as E3 ubiquitin ligases that play important roles in NLR signaling. Pellino3 mediates activation of NOD2 to facilitate gut homeostasis[48] and we now show that Pellino2 mediates activation of NLRP3 and its inflammasome complex. Furthermore, both Pellino proteins target kinases in manifesting these effects with Pellino3 promoting ubiquitination of RIP2 to mediate NOD2 signaling and Pellino2 targeting IRAK1 to facilitate NLRP3 inflammasome activation. Pellino1 also acts as an E3 ubiquitin ligase for RIP1 in the TLR3 and TLR4 pathways[40]. The present study now confirms that all Pellino family members target and ubiquitinate key regulatory kinases in innate immune signaling.

Our data also highlight the intriguing ability of Pellino family members to differentially regulate the same pathway. We have recently shown that Pellino3 can negatively regulate the expression of pro-IL-1β by destabilizing the transcription factor HIF-1α and so suppress the transcription of the *Il1b* gene[49]. We now

**Fig. 8** Pellino2 ubiquitinates IRAK1 and suppress the inhibitory effects of IRAK1 on NLRP3. **a** Immunoblot analysis of myc and IRAK1 in lysates (Input) and immunoprecipitated (IP) myc samples from HEK293T cells transfected with myc-tagged Pellino2 and untagged IRAK1. **b** Immunoblot analysis of HA, IRAK1 and FLAG in lysates (Input) and IP IRAK1 samples from HEK293T cells transfected with FLAG-tagged Pellino2, FLAG-tagged Pellino2 RING mutant, untagged IRAK1, and HA-Ubiquitin. **c, d** Immunoblot analysis of Ubiquitin (short exposure (SE) or long exposure (LE)) and IRAK1 in lysates (Input) and IP IRAK1 samples or IRAK1 and ubq in TUBE-Ubq elution and cell lysates (**d**) from WT and $Peli2^{-/-}$ BMDMs treated with 100 ng/ml of LPS for the indicated times. **e** Immunoblot analysis of NLRP3 and IRAK1 in lysates (Input) and IP IRAK1 samples from WT and Peli2$^{-/-}$ BMDMs treated with 100 ng/ml LPS for the indicated times. **f** ELISA of IL-1β (left panel) and TNF (right panel) in medium from primary WT and $Irak1^{-/-}$ BMDMs treated with 100 ng/ml LPS for 3 h and then with 2.5 mM ATP or 5 mM Nigericin for 1 h. UT, untreated. **g** Immunoblot analysis of IL-1β and Caspase-1 in medium (Sup) and lysates of WT and $Irak1^{-/-}$ BMDMs stimulated with 100 ng/ml LPS for 3 h and 2.5 mM ATP for 1 h. **h** Immunoblot analysis of HA, NLRP3, and IRAK1 in lysates (Input) and IP NLRP3 samples from HEK293T cells transfected with V5-tagged NLRP3, untagged IRAK1, untagged kinase dead IRAK1 (IRAK1-KD), and HA-ubiquitin. β-Actin was used as loading controls. **i** Immunoblot analysis of Ubiquitin, NLRP3, and IRAK1 in lysates (Input) and IP NLRP3 samples from immortalized WT and $Irak1^{-/-}$ BMDMs treated with 100 ng/ml LPS for the indicated times. **j** Immortalized WT and $Irak1^{-/-}$ BMDMs were infected with MSCV as control (Ctrl) or with MSCV containing an expression construct encoding myc-tagged murine Pellino2 (Peli2), Pellino2 RING mutant (RING) or Pellino2 FHA mutant (FHA). Immunoblot analysis of myc and NLRP3 in lysates (Input) and IP myc samples from virus-infected BMDMs treated with 100 ng/ml LPS for indicated times. *$p < 0.05$ (paired, two-tailed Student's $t$-test). Data are biological replicates that are representative of three independent experiments (**a-e, g-j**) or mean ± s.e.m. of three independent experiments (**f**)

# ARTICLE

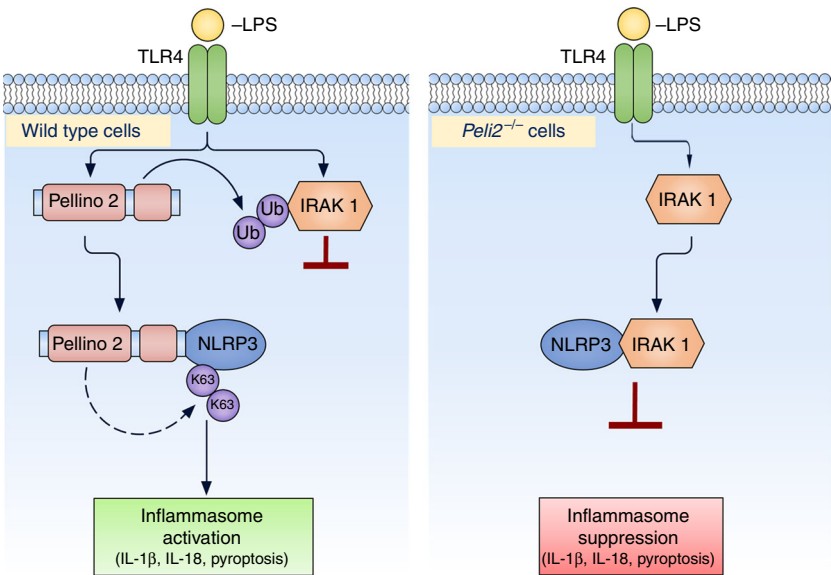

**Fig. 9** Schematic representation of Pellino2 mediated NLRP3 inflammasome priming. In wild type macrophages LPS promotes the association of Pellino2 with NLRP3 and so facilitates ubiquitination of NLRP3. This promotes NLRP3 inflammasome assembly, ASC oligomerization and downstream caspase-1-mediated processing of pro-IL-1β and pro-IL-18 and pyroptosis. Pellino2 also promotes ubiquitination of IRAK1 and so limits the interaction of IRAK1 with NLRP3 and prevents the inhibitory effects of IRAK1 on NLRP3 activation. In *Peli2*$^{-/-}$ macrophages this braking effect of Pellino2 on IRAK1 is removed, allowing for IRAK1 to interact with NLRP3 and suppress downstream activation of the NLRP3 inflammasome

demonstrate that Pellino2 can promote increased production of mature bioactive IL-1β by facilitating activation of the NLRP3 inflammasome. This suggests that Pellino2 may favor the generation of a pro-inflammatory environment whereas Pellino3 seems to temper the inflammatory response.

The involvement of Pellino2 in inflammasome activation appears to be limited to NLRP3. Our data suggests that Pellino2 does not regulate caspase-11 expression or non-canonical inflammasome activation in cells. Interestingly Pellino2 deficiency offers some protection in mice against the lethal effects of LPS and yet such lethality has previously been attributed to caspase-11-induced pyroptosis[58]. While caspase-11 cannot directly cleave pro-IL-1β or pro-IL-18, it can promote indirect formation of the NLRP3 inflammasome by Gasdermin D-mediated membrane pore formation and pyroptosis. Thus, caspase-11 can indirectly induce processing of the precursors of both IL-1β and IL-18[59]. Such activation of NLRP3 may be compromised in Pellino2-deficient mice and underlie the reduced serum levels of IL-1β and increased survival in response to in vivo administered LPS. However lack of Pellino2 does not affect the ability of transfected LPS, a means of activating non-canonical inflammasome activation, to promote secretion of IL-1β from BMDMs. The mechanism by which caspase-11 activates the NLRP3 pathway remains to be fully delineated and the lack of effect of Pellino2 deficiency on this non-canonical pathway may reflect a different activating mechanism than the Pellino2-dependent mechanism employed in the canonical NLRP3 pathway that is triggered by LPS/ATP.

The mediatory role of Pellino2 in NLRP3 activation represents a new mechanistic insight into the molecular basis by which this inflammasome is activated. While Pellino2 is not a player in the activation of NFκB and induction of pro-IL-1β and NLRP3, we now propose a new regulatory pathway that involves Pellino2-dependent ubiquitination of NLRP3. Previous reports have described TRIM31, FBXL2, and MARCH7 to promote ubiquitination and degradation of NLRP3[24–26] with de-ubiquitination by the BRCC3 complex facilitating activation of NLRP3[27–29]. While TRIM31, FBXL2, and MARCH7 are more associated with K48-

linked ubiquitination and degradation of NLRP3, we show that NLRP3 is initially modified with K63-linked polyubiquitin chains, in response to priming signals such as LPS. The loss of Pellino2-dependent ubiquitination of NLRP3 leads to impaired inflammasome activation suggesting K63-linked ubiquitination of NLRP3 to be an important part of the priming process. Indeed MCC950, a highly potent and selective inhibitor of NLRP3 strongly suppresses this priming-induced K63-linked ubiquitination of NLRP3.

Having performed its role as part of the priming process, the ubiquitin chains on NLRP3 presumably need to be removed by BRCC3 to facilitate downstream inflammasome complex formation. While the molecular basis to the positive effects of Pellino2-dependent ubiquitination remains to be delineated, it may facilitate the transient recruitment of an accessory protein(s) that facilitates NLRP3 activation or alternatively impair the binding of an inhibitory protein such as IRAK1, as demonstrated in this study. While our data clearly demonstrate that Pellino2 can promote ubiquitination of NLRP3, we were unable to show, using in vitro ubiquitination assays, that NLRP3 is a direct substrate for Pellino2. This suggests that the ubiquitination is mediated by another E3 ubiquitin ligase or that the Pellino2-induced ubiquitination is dependent on some ancillary protein or process that is triggered by the priming signal. Our data also adds to a growing appreciation of the importance of ubiquitination for the NLRP3 inflammasome and its various constituents. Linear ubiquitination of ASC, by the LUBAC component HOIL-1L, is required for NLRP3 inflammasome assembly[60] and polyubiquitination of pro-IL-1β facilitates its interaction with caspase-1[61], a prologue to assembly of the NLRP3 complex[61, 62]. In addition, the E3 ligases cIAP1 and cIAP2 promote K63-linked polyubiquitination of caspase-1, a pre-requisite for its full activation[63] whereas autophagy targets ubiquitinated pro-caspase-1 and inflammasomes for degradation[64].

Since we were unable to show that Pellino2 directly ubiquitinates NLRP3, we explored the possibility that it could target an intermediate protein to effect such ubiquitination. We have previously shown Pellino2 to interact with IRAK1[65] and the latter can also associate with NLRP3[66, 67] and thus we explored the

regulatory effects of Pellino2 on IRAK1. We now show that Pellino2 mediates LPS-induced ubiquitination of IRAK1 that is associated with impaired binding to NLRP3. Intriguingly we also show that IRAK1 acts to negatively regulate the NLRP3 pathway and this is consistent with reduced binding of Pellino2 to NLRP3 and ubiquitination of the latter. Thus, we now present a novel network for regulating NLRP3 in which Pellino2 acts to promote ubiquitination of NLRP3, an important part of the priming process, and this is counter-regulated by IRAK1 that can bind to NLRP3 to suppress its ubiquitination and activation. IRAK1 kinase activity appears to be very important for mediating these effects since a kinase inactive form of IRAK1 strongly favors ubiquitination of NLRP3. Interestingly, previous studies have suggested a positive role for IRAK1 in acute NLRP3 activation that is triggered by short co-treatment with LPS and ATP that is independent of priming and new protein synthesis[66, 67]. However, we now propose a negative role for IRAK1 in the priming phase of NLRP3 activation and this is consistent with an earlier study showing increased processing of caspase-1 in IRAK1 knockout cells under these conditions[67]. The counter-regulatory roles of Pellino2 and IRAK1 in this pathway highlights the close interplay between the IRAK and Pellino families. Previous reports have described IRAKs as being capable of phosphorylating Pellino proteins to increase their E3 ubiquitin ligase activities. However, it is highly unlikely that IRAK1-induced phosphorylation and activation of Pellino2 is upstream of NLRP3 inflammasome activation since we show enhanced interaction of Pellino2 with NLRP3 and increased ubiquitination and activation of NLRP3 in IRAK1-deficient cells. Furthermore, in the context of LPS signaling, IRAKs are not employed for this purpose and instead TBK1/IKKε have been proposed to act as the activating kinases for Pellino proteins[38]. Instead, we speculate that IRAK1 may promote phosphorylation of NLRP3 to restrict recruitment of Pellino2. Intriguingly, NLRP3 activation is negatively controlled by phosphorylation of its pyrin domain[68]. While the effector kinases were not identified in the latter report, our studies now highlight the importance of IRAK1 as a negative regulator of NLRP3 and as a lead candidate kinase that underlies the inhibitory effects of NLRP3 phosphorylation. We also describe a new regulatory network, involving an E3 ubiquitin ligase and a kinase, that controls NLRP3 activation and ascribe the first physiological role to Pellino2. Such intricate control of this inflammasome likely serves a crucial role in precluding dysregulated NLRP3 activation and uncontrolled inflammatory diseases. Future studies on this pathway are warranted to provide a better understanding of associated disease and to provide important clues for novel interventive strategies.

## Methods

**Mice.** Peli2[−/−] mice were generated by Taconic Artemis using proprietary technology. To generate constitutive Peli2[−/−] mice, mice that were heterozygous for the targeted allele were bred with mice containing a Flpe transgene (C57BL/6-Tg (CAG-Flpe)2Arte). This resulted in the deletion of exon 2–6 and loss of function of the Peli2 gene. The Flpe transgene was removed by breeding the resulting Peli2[+/−] mice with C57BL/6 mice during colony expansion. Mice were genotyped by PCR analysis of DNA isolated from ear punches using primers "a," GCCTCTA-CAGGATGCTCATTT; "b," GGACAGTCATGCTAGTCTGAGG; "c," GAGACTCTGGCTACTCATCC; and "d," CCTTCAGCAAGAGCTGGGGAC. Bone marrow from Irak1[−/−] mice were provided by Prof. Katherine Fitzgerald, the University of Massachusetts Medical School. Genotype of mice was confirmed by western blot for IRAK1 in cultured BMDMs[55]. All animal experiments were performed under licenses of the Health Products Regulatory Authority (HPRA) of Ireland and the UK Home Office with all protocols being approved by the Research Ethics committees of Maynooth University or Queens University Belfast. Sample sizes used are in line with other similar published studies. All animals were used at age 8–12 weeks. Animals were allocated to experimental groups based on genotype, age, and sex. No randomization methods were used. Mice were housed in individually ventilated cages. Mice were culled using cervical dislocation in isolation from any other mice.

**Plasmids and reagents.** The plasmids myc-tagged Pellino2, Pellino2-C335A/338A (Peli2-RING), and Pellino2-T187A/N188A (Peli2-FHA) were generated in house; V5-NLRP3 was donated by Prof. Jae Jung, the University of Southern California. IRAK1 and FLAG-tagged IRAK4 were generated in house. HA-ubiquitin was from Addgene. All antibodies were used at a dilution of 1:1000 unless otherwise stated. Anti-ERK (9101), anti-ERK (9102), anti-p-IκBα (9246), anti-pIKK (14938), anti-p65 (3033) anti-p-P38 (9211), anti-P38 (9212), anti-p-Jnk (9251), anti-Jnk (9252), anti-myc (2276), anti-IRAK1 (4504), anti-IRAK4 (4363), and anti-human caspase 1 p20 (4199) were from CellSignaling; anti-mouse caspase1 p20 (AG-20B-0042-C100) and anti-mouse NLRP3 (AG-20B-0014-C100) was from adipogen; anti-mouse IL1β (AF-401-NA) was from R&D; anti-IκBα (C-21; sc-371), anti-Ub (P4D1; sc-8017), and anti-ASC (sc-514414) was from Santa Cruz; anti-HA (16B12; MMS-101P) was from Covance; anti-K63 ubiquitin (HWA4C4; BML-PW0600) was from Enzo life science; anti-K48 ubiquitin (Apu2; 05-1307) was from Millipore; anti-flag (F3165) and anti-β-actin (AC-15; A 1978) were from Sigma; anti-mouse IRDyeTM 680 (926-68070) and anti-rabbit IRDyeTM 800 (926-32211) were from LI-COR Biosciences; anti-mouse-HRP and anti-rabbit-HRP were from Promega. Anti-rabbit Alexa Fluor 568 (A-11011) was from Invitrogen. Adenosine 5′-triphosphate disodium salt hydrate (ATP) (A1852) and Nigericin (N7143) were from Sigma. LPS (ALX-581-010-L002) was from Enzo. Pam3CSK4, Pam2CSK4, Zymosan, Poly (I:C), Flagellin, Clo75, Clo97, CpG, Alum, and Poly(dA:dT) were from Invivogen. Murine IL-1β was from R&D. Cells were transfected using Lipofectamine 2000 (Invitrogen) according to the manufacturer's instructions. LDH cytotoxicity assay kit (61780) was from Promega.

**Cell culture.** HEK293T (ATCC CRL-11268) were cultured in Dulbecco's modified Eagle's medium supplemented with 10% (v/v) fetal bovine serum, 100 U/ml penicillin and 100 μg/ml streptomycin. THP1 cells were cultured in RPMI-1640 medium supplemented with 10% (v/v) fetal bovine serum, 100 U/ml penicillin and 100 μg/ml streptomycin. All cell lines were obtained from the ATCC and have been previously validated using the STR method by the ATCC. Cell lines were not tested for mycoplasma contamination. For isolation of BMDMs, tibias and femurs were removed from WT and Peli2[−/−] mice by sterile techniques and bone marrow was flushed with fresh RPMI-1640 plus GlutaMAX-I medium using a 27[1/4] gage needle. Cells were plated in medium supplemented with 10% (v/v) conditioned medium of L929 mouse fibroblasts and were maintained for 6 days at 37 °C in a humidified atmosphere of 5% $CO_2$. Medium was replaced every 2 days. Peritoneal macrophages were harvested by injecting sterile PBS containing 10% (v/v) fetal bovine serum (5 ml) into the peritoneal cavity. The peritoneal exudate was withdrawn and centrifuged at 1500 g for 5 min. Pelleted peritoneal cells were resuspended in RPMI containing 10% (v/v) fetal bovine serum and cultured at 37 °C prior to indicated stimulations. Immortalized WT, Irak1[−/−], and Irak4[−/−] BMDMs (iBMDMS) were provided by Prof. Katherine Fitzgerald, the University of Massachusetts Medical School. iBMDMs were cultured and passaged every 3 days in fresh RPMI-1640 plus GlutaMAX-I medium.

**ELISA.** Primary BMDMs were seeded (1 × 10[6] cells per ml; 200 μl/well) in 96-well plates and allowed to rest for 24 h. Cells were then stimulated with the indicated ligands. Conditioned media was collected at the indicated time points and IL-1β, IL-18, TNF, IL-6, RANTES, and CXCL1 were quantified by sandwich ELISA (R&D Systems).

**Cytotoxicity assay.** Conditioned medium from treated BMDMs was assessed for LDH release using the CytoTox96 non-radioactive cytotoxicity assay (Promega) as per manufacturer's instructions.

**Inflammasome activation assays.** Cells were primed with LPS (100 ng/ml) from E. coli serotype EH100 (ra) TLRgrade for 3 h followed by stimulation with the inflammasome activators: adenosine 5′-triphosphate disodium salt hydrate (ATP) (5 mM) for 1 h, Poly (dA:dT) (1 μg/ml) transfected with Lipofectamine 2000 (Invitrogen) for 6 h or Nigericin (5 μM) for 1 h. For noncanonical inflammasome activation, cells were primed with 100 ng/ml Pam3CSK4 (Invivogen) for 3 h, followed by transfection of LPS (2 μg/ml) using Lipofectamine 2000 for 6 h or primed with LPS for 3 h and stimulated with CTB (20 μg/ml) for 16 h.

**Immunoblotting and immunoprecipitation.** Primary BMDMs from WT and Peli2[−/−] mice were cultured in 12-well plates (1 × 10[6] cells per ml; 1 ml) or 10 cm dishes (2 × 10[6] cells per ml; 10 ml). HEK293T cells (2.5 × 10[5] cells per ml; 3 ml) were grown in 6-well plates and where indicated were transfected with the appropriate expression constructs. For whole cell lysate analysis, cells were lysed in NP-40 lysis buffer (50 mM Tris-HCl, pH 7.4, containing 150 mM NaCl, 1% (w/v) IgePal, 50 mM NaF, 1 mM Na3VO4, 1 mM dithiothreitol, 1 mM phenylmethylsulfonyl fluoride and complete protease inhibitor mixture (Roche)). For co-immunoprecipitation, cells were treated as indicated and then collected in 200 μl 0.5 × NP-40 lysis buffer, followed by incubation for 30 min at 4 °C. Cell lysates were then incubated with the appropriate antibody and an aliquot (50 μl) of protein A–protein G–agarose was added to each sample, followed by incubation overnight at 4 °C. Immunoprecipitates were collected by centrifugation for 1 min at 1000 g at 4 °C and the beads were then washed four times with 1 ml of NP-40 lysis buffer

(without Na3VO4, dithiothreitol, phenylmethylsulfonyl fluoride, or protease-inhibitor "cocktail"). An aliquot (40 μl) of SDS-PAGE sample buffer (62.5 mM TrisHCl, pH 6.8, 10% (w/v) glycerol, 2% (w/v) SDS, 0.7 M β-mercaptoethanol, and 0.001% (w/v) bromophenol blue) was added to the beads. Samples were resolved by SDS-PAGE, transferred to nitrocellulose membranes and analyzed by immunoblot with the following antibodies as appropriate: anti-NLRP3, anti-IRAK1, anti-myc, and anti-FLAG. Immunoreactivity was visualized by the Odyssey Imaging System (LICOR Biosciences) or enhanced chemiluminescence. For experiments assessing the ubiquitination status of NLRP3 or IRAK1, cells were collected in 200 μl RIPA buffer (25 mM Tris-HCL, 150 mM NaCl, 1% (w/v) IgePal, 1% (w/v) sodium deoxycholate). Cell lysates were treated with 1% (w/v) SDS and heated to 95 °C for 5 min to dissociate NLRP3 or IRAK1 from any associated proteins. Lysates were then diluted 10 fold in RIPA buffer before immunoprecipitation. Immunoprecipitated samples were analyzed by immunoblot with anti-ubiquitin (P4D1; sc-8017; Santa Cruz). For immunoblotting of IL-1β and Caspase-1 in cell supernatants, conditioned medium was collected and filtered using filter spin columns to reduce salt and remove abundant serum proteins. Filtrates were added to 4 × SDS-PAGE sample buffer and resolved by SDS-PAGE for immunoblot analysis.

**Tandem ubiquitin binding entity analysis.** Primary BMDMs from WT and *Peli2*$^{-/-}$ mice were cultured in 10 cm dishes (2 × 10$^6$ cells per ml; 10 ml). For enrichment of ubiquitinated proteins, Ubq-TUBE-Agarose (UM-401) or K63-TUBE-FLAG (UM-604) or K48-TUBE-FLAG (UM-605) (Life Sensors) were used according to manufacturer instructions. Briefly, cells were lysed in TUBE lysis buffer (100 mM Tris HCL pH 7.5, 150 mM NaCl, 5 mM EDTA, 1% NP-40, 0.5% Triton) containing 1,10-phenanthroline (5 mM), PR-619 (100 μM) N-Ethylmalamide (5 mM) and 20 μl of Ubq-TUBE agarose or 500 nM of K63-TUBE-FLAG or K48-TUBE-FLAG. Cell lysates were then incubated for 1 h on ice to allow TUBE binding. For enrichment of FLAG-tagged TUBEs, M2-FLAG affinity gel (A220, Sigma) was equilibrated in TBST for 5 min then added to samples (20 μl/sample) and incubated at 4 °C for 2 h with rotation. TUBE-enriched FLAG affinity gel was then collected by centrifugation for 5 min at 5000 rpm at 4 °C and washed three times with 1 ml of wash buffer (TUBE lysis buffer without Triton). An aliquot (25 μl) of SDS-PAGE sample buffer (62.5 mM TrisHCl, pH 6.8, 10% (w/v) glycerol, 2% (w/v) SDS, 0.7 M β-mercaptoethanol and 0.001% (w/v) bromophenol blue) was added to the beads. Samples were resolved by SDS-PAGE, transferred to nitrocellulose membranes and analyzed by immunoblot with the following antibodies as appropriate: anti-NLRP3, anti-IRAK1, anti-K63-linked ubiquitin, anti-K48 linked ubiquitin, and anti-ubiquitin. Immunoreactivity was visualized by the Odyssey Imaging System (LICOR Biosciences).

**Real time PCR analysis.** Total RNA was extracted from tissues or cells using Trizol (Invitrogen). cDNA was generated from 2 μg RNA using MMLV Reverse Transcriptase (Bioscript Bioline) and real-time PCR analyses were performed with SensiMix SYBR Master mix (Bioline) using an Applied Biosystems StepOnePlus™ Real-Time PCR System according to the manufacturer's instructions. The abundance of each mRNA was normalized relative to PCR of the housekeeping gene Hypoxanthine-guanine phosphoribosyltransferase (Hprt). Mouse Il1β, forward, CGGCACACCCACCCTG, and reverse, AAACCGTTTTTCCATCTTCTTCT; mouse Il6, forward, ACAACCACGGCCTTCCCTAC, and reverse, TCCACGATTTCCCAGAGAACA; The abundance of each mRNA was normalized relative to PCR of the housekeeping gene Hprt with the following primers: forward, GTCCCAGCGTCGTGATTAGC, and reverse, TGGCCTCCCATCTCCTTCA.

**LPS septic shock model.** Mice aged 8–12 weeks were administered LPS 20 mg/kg by intraperitoneal injection. Serum was collected after 6 h and various cytokine assayed by ELISA. For survival analysis, mice were administered LPS 50 mg/kg intraperitoneally and monitored every 6 h for 72 h.

**P. aeruginosa infection.** Male WT and *Peli2*$^{-/-}$ mice were inoculated intraperitoneally with *P. aeruginosa* (PAO1 strain, kindly provided by Prof Scott Bell, the University of Queensland) (100 μl; 1.5 × 10$^7$ CFU), suspended in sterile endotoxin-free PBS, or sham inoculated with sterile endotoxin-free PBS. At 10 h post-inoculation the animals were killed by cervical dislocation. Peritoneal lavage was collected using 5 ml of ice cold endotoxin-free PBS, centrifuged at 1000 g for 10 min and the supernatants assayed for levels of IL-1β, IL-18, and IL-6. For cell-based infections, bacteria were grown for 18 h at 37 °C in LB Broth. Bacteria were then diluted 1/5 in LB broth and cultured for a further 2 h. Indicated cells were infected at a multiplicity of infection (MOI) of 100:1 for 6 h. Cell supernatants and lysates were then collected for ELISA and western blot analysis.

**ASC oligomerization and fluorescence microscopy.** ASC oligomers were analyzed as previously described[54]. Cells were thus treated with LPS with or without ATP or Nigericin for the indicated times. Cells were then washed in PBS and lysed in NP-40 buffer on ice. NP-40 soluble fractions were retained and insoluble pellets were washed twice in PBS. Insoluble pellets were then reconstituted in PBS and cross-linked in 2 mM Suberic acid for 1 h at room temperature with gentle agitation. Cross-linked pellets were washed twice in PBS and lysed in 2 × sample buffer for western blot analysis of ASC oligomers. Coverslips were coated with BMDMs

for fluorescence microscopic analysis of ASC specks. Cell were fixed for 5 min with methanol (−20 °C) and washed three times with PBS. Samples were incubated with anti-rabbit ASC (1:200) overnight, followed by incubation with anti-rabbit Alexa Fluor 568 (5 μg/ml) and mounting in DAPI-containing media. Images were captured using the ×20 objective of a fluorescent microscope. Image analysis was performed using Olympus cellSens Dimension 1.9 software

**Mitochondrial ROS.** Primary BMDMs were treated as indicated and assayed for mROS production using Mitosox (LifeTechnologies) according to the manufacturer's instructions. Briefly, cells were washed twice in PBS and then incubated with 2.5 mM Mitosox in PBS for 15 min at 37 °C. Cells were then washed twice in cold PBS (1 ml) followed by flow cytometric analysis of Mitosox-reactive cells using an AcuriC6 flow cytometer and Flowjo software

**Preparation and transfection of siRNA.** Human Pellino2-specific siRNAs were from the LifeTechnologies (AM16708, 133379). siRNA was delivered to THP1 cells via transfection with Lipfectamine2000 according to the manufacturer's instructions and allowed to recover for 48 h prior to experiments.

**Retroviral rescue assay.** Myc-tagged murine Pellino2 (mPeli2), Pellino2-C337A/334A (mPeli2-RING), and Pellino2-R106A/R136A (mPel2-FHA) were sub-cloned into the retroviral MSCV2.2-IRES-GFP vector. The empty MSCV vector (as control) or MSCV containing the indicated Pellino2 construct (10 μg) was co-transfected with the packaging vector ψ (5 μg) and VSV-g envelope vector (5 μg) into a T75 cm$^2$ flask of HEK293T cells in DMEM medium (10 ml). After 48 h, the retrovirus-containing medium was collected. WT and Pellino2-deficient BMDMs were then plated with virus containing medium (ratio of 1:1 (v:v)) in 12-well plates for ELISA/cell lysate analysis or 10 cm$^2$ dishes for co-immunoprecipitation analysis. Retrovirus-infected cells were incubated at 37 °C for 24 h prior to experiments.

**Densitometry analysis.** Densitometry analysis was performed using Licor Odyssey and Gel QUANT software.

**Statistical analysis.** Student's *t*-test and two-way Mantel-Cox were used, where appropriate, for statistical analysis. 3–12 mice were used per experiment, sufficient to calculate statistical significance and in line with similar studies published in the literature. No animals or samples were excluded from the analysis

**Data availability.** All source data is available upon request.

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

## Acknowledgements

This publication has emanated from research conducted with the financial support of the Science Foundation Ireland (SFI) under Grant Numbers SFI/16/IA/4622 and SFI/12/IA/1736.

## Author contributions

F.H. and R.B. developed the concept, designed, and performed experiments, analyzed data and prepared the figures; F.H. also assisted with writing the manuscript. R.J. designed and performed experiments and analyzed data; N.D. assisted with experiments, interpreted experimental data and advised on manuscript preparation. B.W. generated expression constructs and managed the breeding program for mice. S.Y generated Pellino2 expression constructs. A.V.D. and R.J.I. designed, performed, analyzed, and

supervised the *P. aeruginosa* studies. P.N.M. conceived the study, supervised the overall project, analyzed data, and wrote the manuscript.

## Additional information

**Competing interests:** The authors declare no competing interests.

