## [Peer Review File · Nature Communications]

Reviewers' comments:

Reviewer #1 (Inflammasome, sterile inflammation)(Remarks to the Author):

"The E3 ubiquitin ligase Pellino2 mediates priming of the NLRP3 inflammasome" (NCOMMS-17-12750-T) by Moynagh et. al.

Humphries et al. analyze the function of the E3 ubiquitin ligase Pellino2 in innate immunity using Pellino2 deficient mice (Peli2^{-/-}). Given that other members of the Pellino family play important roles in inflammatory signaling, the authors start out by measuring cytokine and chemokine secretion in BMDMs upon stimulation with different TLR agonists or zymosan. Intriguingly, specifically IL-1 β secretion in response to LPS is impaired in Peli2^{-/-} BMDMs. The authors convincingly show that Pellino2 is not required for the NF- κ B mediated 'signal 1' of inflammasome priming, but for the subsequent step of inflammasome activation. The role of Pellino2 is specific for the Nlrp3 inflammasome, because the responses to stimuli for the AIM2, Nlrp4 or the non-canonical inflammasome are unaltered in Peli2^{-/-} cells. The observations are complemented by data from in vivo experiments. In a model of sterile sepsis, Peli2^{-/-} mice show less IL-1 β in the serum and they survive better after intraperitoneal LPS injection. Moreover, in Peli2^{-/-} mice that are infected with *P. aeruginosa*, IL-1 β and inflammasome dependent IL-18 levels are likewise reduced. These are important genetic data that underscore the role of Pellino2 for inflammasome activation in relevant disease or infection models.

From structure-function analyses the authors subsequently conclude that both Pellino2's RING and FHA domain are necessary for full activation of the Nlrp3 inflammasome and less ASC filaments (as a marker for inflammasome activation) are detected in BMDMs lacking Pellino2. The authors observe that upon LPS stimulation, Nlrp3 is ubiquitinated in a Pellino2-dependent manner, although the biological function of this ubiquitination remains unclear (or whether there is a function to it at all). Although Nlrp3 and Pellino2 co-immunoprecipitate and despite Pellino2 being an E3 ubiquitin ligase, Pellino2 is not sufficient to ubiquitinate Nlrp3, making it unlikely that Nlrp3 is a direct substrate of Pellino2. Further data presented in this manuscript support a model according to which Pellino2 and IRAK1 somehow control the licensing of Nlrp3 by antagonistically regulating Nlrp3's ubiquitination status.

Taken together the study by Humphries et al. demonstrates a clear physiological function of Pellino2 and provides initial insights into the potential mechanistic underpinnings. The study appears technically sound and includes high-quality data that are of interest to immunologists. The phenotype seems robust and the in vivo data clearly show physiological relevance for Pellino2 function in inflammatory signals. However, the molecular events initiated by Pellino2 to regulate Nlrp3, are not entirely clear and addressing the following points could help to further improve the manuscript:

Specific point:

1. In general, MW markers are frequently missing in the immunoblots and should be added to help with interpretation.
2. In fig. 5c the meaning of the bands in the caspase-1 p20 immunoblot remain unclear and the authors conclusion from the data is also unclear. Caspase-1 seems to be cleaved under all condition. In the presence of wild type Pellino2, the cleavage fragment seems to be larger than in the absence of Pellino2 or with the Pellino2 mutants. Further comments from the authors would be helpful.
3. If the authors conclude that the signal for processed caspase-1 is reduced in the absence of wild type Pellino2, a quantitation from several replicate experiments should be provided. In addition, blotting for IL-1 β in the supernatants would be helpful to interpret the data.

4. The immunoblot data from MCC950-treated samples (figs. 7d and e) are somewhat confusing. If MCC950 inhibits ubiquitination of NLRP3, why is there consistently more ubiquitinated NLRP3 in unstimulated, MCC950-treated cells? Also, how would the authors explain that in fig. 7d ubiquitination of NLRP3 is not reduced after adding ATP? The conclusion that "targeting of NLRP3 ubiquitination may represent a plausible mechanism of action" for MCC950 seems rather speculative based on the data provided in this manuscript.
5. In fig. 8b, a second band appears in the anti-FLAG (Pellino2) blot only in the presence of IRAK1. How do the authors explain this band? Is it phosphorylated Pellino2 (does the band disappear with phosphatase treatment)? Is this band also seen if IRAK1-KD is used instead of wild type IRAK1? Is there a functional relevance of this potential modification?
6. In the same panel, a band in the IRAK1 blot (input) disappears only when wild type Pellino2 is co-expressed. How do the authors explain these findings? Potentially, these results could give further insights into the somewhat unclear mechanism of the Pellino2 / IRAK1 / Nlrp3 interaction.
7. The authors propose a model according to which Pellino2 ubiquitinates IRAK1, thus favoring IRAK1 dissociation from NLRP3 (last paragraph of the results section). However, neither do they show that IRAK1 is a substrate of Pellino2 nor that ubiquitination of IRAK1 is indeed the cause for less interaction of IRAK1 with NLRP3. The relevant section(s) should be rephrased to prevent conclusions that are not supported by the data.
8. How do the authors explain the results of the IRAK1 blot (input) in fig. 8g? Was IRAK1-KD expressed?
9. A schematic diagram of the proposed model would enhance readability of the manuscript.

Minor concerns:

- a. In several immunoblots it is not clear which bands are the bands of interest (e.g. bands in the IRAK1 blots in fig 8d, NLRP3 blots in fig. 7e and f, etc.)
- b. The overall quality of some of the immunoblots shown is in my opinion suboptimal. For example, the ubiquitin blot in Fig. 7b is overexposed, the signal of the caspase-1 p20 blot in fig 8f is very weak, protein transfer for the ubiquitin blot in fig. 8c seems uneven.
- c. The graphs in some figure panels are not correctly or insufficiently labeled (e.g. figs. 4b, 4d, etc.).
- d. Font size and formatting of the figures are inconsistent, which makes reading more difficult.
- e. Names of genes or proteins should be used consistently throughout the manuscript (e.g. caspase 1 vs. caspase1 vs. caspase-1).
- f. Readability would be improved by briefly explaining the RING and FHA mutants in the results section in the context of figure 5.

Reviewer #2 (NLRP3, inflammasome)(Remarks to the Author):

In the manuscript by Humphries and colleagues it is demonstrated that the E3 ubiquitin ligase Pellino2 is involved in NLRP3 inflammasome activation. Myeloid cells from Pellino2-KO mice display impaired NLRP3 response following TLR priming, NLRP3 activation and NLRP3 responses towards bacteria. It is demonstrated that Pellino2 promotes ubiquitination of NLRP3 during the priming phase and that IRAK1 and Pellino2 have opposing effects.

The manuscript describes a novel pathway that regulates NLRP3 activation. The work is of interest to the innate immunity community and addresses an important signaling pathway.

A few concerns should be addressed before publication.

Major:

Figure 5:

The Western blot analysis shows only minor differences in the amount of Caspase p20 between the LPS+ATP stimulated conditions. Especially there seems to be no difference in the amount of p20 between the WT and Peli2-KO BMDMs.

Figure 7(a,b,d,e):

Show size markers for Western blots and check whether the ubiquitin signal only appears at a higher molecular weight than NLRP3. If ubiquitin signal is detected at a lower molecular weight than NLRP3 this indicates that NLRP3 co-immunoprecipitates with other ubiquitinated proteins and a definite answer about the ubiquitination of NLRP3 is not possible.

Please provide uncropped images of NLRP3 to make sure that ubiquitinated NLRP3 is detected at a higher molecular weight compared to unmodified NLRP3.

As an alternative, the authors could perform an IP for ubiquitin itself and stain the Western-blot for NLRP3 (see Emmerich and Cohen, 2015).

Figure 7:

The authors should check the ubiquitin linkage type resulting from Pellino2. Please use linkage type specific antibodies or TUBEs to check for K48 and K63 ubiquitination. The activating mechanism of ubiquitination would be more convincing in case NLRP3 would be modified with an atypical linkage type, as a K48 ubiquitination should only result in enhanced degradation of NLRP3.

Figure 8e:

Show TNF α secretion as a priming control for the experiment.

Figure 8(b,c,g,h):

Show size markers for Western blots and check whether the ubiquitin signal only appears at a higher molecular weight as the precipitated proteins (NLRP3/IRAK1). If ubiquitin signal is detected at a lower molecular size than the target protein, this indicates that NLRP3/IRAK1 co-IP with other ubiquitinated proteins and a definite answer about the ubiquitination of these proteins is not possible.

Minor:

Page 4:

The authors state that there is no understanding of E3 ligases which ubiquitinate NLRP3. However, previous studies identified already TRIM31 (Song, 2016, Nat. Comm.), FXBL2 (Han et al., 2015, JBC) and March7 (Yan et al., 2015, Cell) as NLRP3 E3 ligases regulating NLRP3. Please include these Papers in your discussion.

Page 5/ Figure 1f:

The authors state that the levels of mature IL-1 β were strongly reduced in the Pellino2-KO. The secreted IL-1 β might be as well pro-IL1 β from dying cells. Especially since the amount of secreted IL-1 β is not altered upon LPS stimulation in the rest of the study.

Page 10:

The authors state that NLRP3 might be a direct target of Pellino3. This is supposed to be Pellino2.

Figure 4b:
The figure is not labeled.

Figure 8b:
Label the figure with "IP IRAK1" and not just "IP IRAK"

Reviewer #3 (Caspase, inflammasome)(Remarks to the Author):

This manuscript reports that Pellino-2 is required for efficient LPS-induced NLRP3 ubiquitylation, and as a consequence, downstream ASC/Caspase-1 activation and IL-1 β processing that results following stimulation of cells with a NLRP3 activator, such as ATP or nigericin. The authors also implicate IRAK1 as a negative regulator of NLRP3 priming, in that IRAK1 deficiency reportedly enhanced NLRP3/Pellino-2 interactions, NLRP3 modification with ubiquitin, and hence NLRP3 activity. Although the findings are certainly novel, I have several issues with contradictory or inconsistent data, and hence I am not convinced at this stage that some of the conclusions are warranted – as outlined in my specific comments below. In particular, much of the mechanistic analysis (Figures 7/8) is unclear/contradictory and the results very difficult to interpret. At this stage, I would be hesitant to include these findings unless the authors can robustly document the reproducibility and validity of the experiments/findings (see below).

Comments:

1. Without any molecular weight markers, it is difficult to ascertain the accuracy of the Western blots presented in all figures. Please add these to all blots.
2. Please clarify in the figure legends on how many separate occasions the in vivo experimental results were verified (e.g. Figure 3, figure 4). Why are there different numbers of mice (data points) for each cytokine measurement in Figures 3 and 4 (presumably all cytokines were assessed for each mouse so there should be an equal number of dots for each analysis)?
3. Figure 1e. Why doesn't LPS result in IFN β production?
4. Figure 1f. Wildtype BMDMs should not respond to LPS to cause IL-1 β secretion (i.e. LPS primes/induces precursor IL-1 β production, but a separate inflammasome activator, signal 2, is normally required for its activation and secretion). Are the authors using ultrapure LPS or LPS that is contaminated with other bacterial ligands/PAMPs? Why do some assays show robust IL-1 secretion with LPS alone (e.g. figure 1 [over 2000-6000pg/ml], figure 2k, figure 7c LPS with MCC930 treatment [very strange]), while others show none (e.g. Figure 2a/e, supp fig 4 etc)?
5. Page 6. The authors write "Given that Pellino2 is required for LPS induced production of mature IL-1 β but is redundant for the initial expression of pro-IL-1 β ". No data is provided prior to this statement that shows Pellino2 being required for IL-1 β maturation (i.e. Western blots showing cleavage to the active p17 fragment, or lack thereof in Pellino2 KOs).
6. Figure 2. The error bars (for figure 2 ELISAs where experimental triplicates are shown) are meaningless as they just demonstrate that the authors know how to pipette. The authors should show true biological replicates or pool the independent experiments and, in either case, present each experiment/mouse as a dot point – this would allow readers to accurately assess the true biological

variability within experiments and between genotypes. I strongly recommend that this methodology be applied to all figures, particularly where technical repeats are currently shown.

7. Figure 2d/h. In my opinion these are the critical experiments as they detect mature IL-1 and caspase-1 in WT and Pell2 KO cells. However, the Pell2 KO lanes containing the pertinent treatments (nigericin/atp) appear under-loaded relative to the similarly treated WT controls (i.e. cell lysate actin, caspase-1 and pro-IL-1b levels are all reduced compared to WT cells). Therefore it is not surprising less cleaved caspase-1 and IL-1 are detected in the supernatants in the Pell2 KO cells. Could the authors please provide better quality Westerns from independent experiments where the loading does not cloud the interpretation of the data. Detailed ATP/Nigericin dose curves and kinetics of IL-1 and caspase-1 maturation/secretion between WT and Pell2 KO cells would be helpful data that would strengthen the authors findings and conclusions. A particulate NLRP3 activator (such as Alum) should also be tested to define how important Pell2 is for both soluble and particulate NLRP3 stimuli.

8. Figure 2k/l. The results are confusing. Figure 2k shows LPS alone induces secretion of IL-1 (which it shouldn't) and that the addition of ATP or Nigericin treatment has no effect (which it should), while in contrast, figure 2l shows that only LPS/ATP (not LPS alone) cause maturation and secretion of IL-1 and caspase-1? Figure 2l contains ATP and nigericin treatments, while figure 2l only shows ATP treatments? Please clarify.

9. Supp. Figure 4. Transfected LPS activates caspase-11, which in turn activates NLRP3-caspase-1 to cause IL-1 maturation and secretion. Wouldn't the authors' model predict that, contrary to results presented, transfected LPS (which specifically activates NLRP3) should result in reduced IL-1 secretion in Pell2 KO cells? The in vivo data suggests this should be the case (Figure 3) as LPS lethality is largely dependent on caspase-11.

10. Figure 4B. Western blot needs labeling.

11. Figure 4f. Pell2 proIL1 is reduced, suggesting this is the reason for reduced cleavage. The caspase-1 p20 blot is of very poor quality.

12. Figure 5C. The caspase-1 p20 fragment is running at different sizes (e.g. WT vs Pell2 KO controls) and differences in cleavage (e.g. band intensity) are not convincing. Please show the other two independent experiments to help convince readers of differences observed. Blots examining IL-1 cleavage and secretion into the supernatant are lacking, which is strange as the authors have blotted lysates for pro-IL-1 in this experiment.

13. Figure 6D. Levels of soluble ASC are lower in Pell2 KO cells – which makes the data difficult to interpret (i.e. loading appears less than WT cells). The ASC blot (soluble) is also overexposed which probably means the loading differences are even greater than what is already apparent.

14. Figure 7a/b/d/e. This data is difficult to interpret without molecular weight markers to indicate the sizes of the ubiquitin laddering. Figure 7a is very convincing whereas figure 7b is less so (i.e. in figure 7a there appears to be a complete lack of NLRP3 ubiquitination in Pell2 KO cells after 3 hrs of LPS treatment, whereas in 7b after 3 hrs of LPS treatment the ubiquitination of NLRP3 in Pell2 KO cells, although perhaps somewhat reduced, is still quite apparent). Which data set is representative of Pell2 KO cells? I would like to see several independent experiments to gauge the significance/reproducibility of this data.

15. Figure 7d/e. Why does MCC950 alone result in strong NLRP3 ubiquitination (comparable to control LPS treated cells) and what does this mean given that in this scenario LPS treatment in the presence of MCC950 appears to actively trigger the de-ubiquitination of NLRP3 (particularly as MCC950 potentially inhibits NLRP3 activation)? Why does NLRP3 ubiquitination disappear after 3 hrs of LPS treatment in the control treated cells (in contrast to Figure 7a/b), which is often when the NLRP3 activating stimulus

(e.g. ATP) is added to cells which then reportedly triggers NLRP3 deubiquitylation?

16. Figure 7f. Is it expected that Pell2 associates with NLRP3 after 0.5 hr of LPS treatment and that this association is then lost after 2 hrs of LPS treatment, as the data suggests (particularly as NLRP3 ubiquitination, according to Figure 7b, is only strongly detected at 3 hrs [although the data in 7e suggests it is ubiquitinated within 1 hr and disappears at 3 h – which result is representative?])? How robust are these results (please show the additional 2 experimental repeats of this data; 7f, 7b, 7e)?

17. Figure 8c. From this data, the authors conclude that “the greatly reduced ubiquitination of IRAK1 in Peli2^{-/-} cells still permitted degradation of IRAK1, albeit with slightly delayed kinetics and more residual IRAK1 at later time points than in WT cells.” I am not convinced that the data shows this. If anything, to my eye there appears to be more residual IRAK1 in WT cells (although levels are almost undetectable). Please provide the independent experiments of this data to demonstrate the robustness of this data and the conclusions that have been drawn from it.

18. Figure 8d. How can IRAK1 associate with NLRP3 after one, two and three hours of LPS stimulation when it is almost completely degraded within 15 minutes of LPS stimulation?

19. Figure 8e/f. These experiments are performed with immortalized BMDMs which have been known to exhibit variable levels of ASC. Please blot and compare levels of total ASC and caspase-1 in WT, IRAK1 and IRAK4 iBMDMs (control blots that are currently lacking in 8f). This is important as previous work using primary BMDMs from IRAK1 deficient mice have reported that IRAK1 is required for efficient NLRP3 responses, including ASC oligomerisation (e.g. PMID: 24043892). The authors should examine primary macrophages derived from IRAK1 deficient mice to ensure their results using iBMDMs are not just a consequence of immortalization and/or genetic drift.

20. Figure 8h. Why isn't NLRP3 ubiquitylated in response to LPS treatment in WT cells (in contrast to the data provided in figure 7). Again, how do the authors explain robust ubiquitylation at 1.5h (this time in IRAK1 KO cells) and then a loss of this signal at 3 hr (which is normally when the NLRP3 signal 2 is provided)?

21. Figure 8i. Please provide the independent experiments and a proper time course of NLRP3 binding to Pell2 (not just the 30 min LPS treatment shown) to define its association with NLRP3 when NLRP3 ubiquitylation is observed.

Dear Editor

Re: Re-Submission of NCOMMS-17-12750-T

Please find submitted a revised Manuscript entitled “**The E3 ubiquitin ligase Pellino2 mediates priming of the NLRP3 inflammasome**” (NCOMMS-17-12750-T) for consideration for publication in *Nature Communications*. We appreciate your invitation to revise our manuscript in response to the reviewer’s comments and we believe that the revised manuscript, complete with new data, now addresses all of the substantive issues raised. We appreciate the constructive feedback from each of the reviewers and feel that their comments and suggestions have greatly enhanced the quality of the manuscript. We hope that you now consider the manuscript worthy of publication in *Nature Communications*.

Responses to the individual points raised by each of the reviewers are now detailed below:

Reviewer #1:

We welcome the reviewer’s comments that the study “*demonstrates a clear physiological function of Pellino2 and provides initial insights into the potential mechanistic underpinnings. The study appears technically sound and includes high-quality data that are of interest to immunologists. The phenotype seems robust and the in vivo data clearly show physiological relevance for Pellino2 function in inflammatory signals.*”. We are also happy to address the specific points raised by the reviewer.

Specific points

1. In general, MW markers are frequently missing in the immunoblots and should be added to help with interpretation.

Response: The mobilities of MW markers are now indicated in all immunoblots.

2. In fig. 5c the meaning of the bands in the caspase-1 p20 immunoblot remain unclear and the authors conclusion from the data is also unclear. Caspase-1 seems to be cleaved under all condition. In the presence of wild type Pellino2, the cleavage fragment seems to be larger than in the absence of Pellino2 or with the Pellino2 mutants. Further comments from the authors would be helpful.

Response: The bands in the caspase-1 p20 blots represented migration of the processed (p20) form of caspase-1 with the apparent different sizes being due to slight irregular running of samples in some individual lanes. To address these concerns we have now included immunoblots from another replicate of this experiment in Fig. 5c. The data clearly show reduced LPS/ATP-induced processing of pro-caspase-1 to its mature p20 form in Pellino2-deficient cells with this deficiency in processing being rescued by reconstitution of Pellino2-deficient cells with wild type Pellino2 but not

with forms of Pellino2 containing mutations in its RING-like or FHA domains.

3. If the authors conclude that the signal for processed caspase-1 is reduced in the absence of wild type Pellino2, a quantitation from several replicate experiments should be provided. In addition, blotting for IL-1beta in the supernatants would be helpful to interpret the data.

Response: As requested, the reproducibility of these studies is now demonstrated by new data (**Figure 5d**) that represent densitometric analysis of caspase-1 p20 across all replicates of this experiment. Furthermore, as requested by this reviewer, such Pellino2-dependent processing of pro-caspase-1 to its mature active form is further confirmed by additional data showing reduced processing of pro-IL-1 β into its mature p17 form in Pellino2-deficient cells (**Figures 5c and 5d**).

4. The immunoblot data from MCC950-treated samples (figs. 7d and e) are somewhat confusing. If MCC950 inhibits ubiquitination of NLRP3, why is there consistently more ubiquitinated NLRP3 in unstimulated, MCC950-treated cells? Also, how would the authors explain that in fig. 7d ubiquitination of NLRP3 is not reduced after adding ATP? The conclusion that “targeting of NLRP3 ubiquitination may represent a plausible mechanism of action” for MCC950 seems rather speculative based on the data provided in this manuscript.

Response: The reviewer raises valid points in relation to the complexity of ubiquitination patterns of NLRP3 that are observed in response to MCC950. In order to provide a better understanding of the mechanism underlying this complexity we have performed more detailed studies to characterize the linkage type of the polyubiquitin chains that are attached to NLRP3 and regulated by MCC950. As indicated by the reviewer, MCC950 inhibits the LPS-induced ubiquitination of NLRP3 whilst also promoting some basal ubiquitination of NLRP3 in unstimulated cells (see **Figure 7e**). However we now include new data from the use of K63-specific TUBEs to isolate K63-linked ubiquitinated proteins to show that LPS promotes strong and time-dependent ubiquitination of NLRP3 and this is fully abrogated in cells that are pre-treated with MCC950 (see new **Figure 7f**). Notably MCC950 alone failed to induce any K63-linked ubiquitination of NLRP3 suggesting that the basal ubiquitination of NLRP3 observed in response to MCC950 is due to ubiquitin chains with different linkages. We present data below showing that MCC950 alone induces time-dependent K48-linked ubiquitination of NLRP3 (Figure I). We have not included this figure in the paper since we hope that it will form part of a future and more in depth investigation of the mechanistic basis to the inhibitory effects of MCC950. We are also keen to ensure that the emphasis of the manuscript remains on Pellino2 and its functions and avoid the paper becoming fragmented and losing focus. These data are consistent with a model in which LPS promotes transient K63-linked ubiquitination of NLRP3 that is associated with NLRP3 activation and this is countered by the inhibitory effects of MCC950 that opposes the K63-linked

ubiquitination of NLRP3 and instead biases towards its K48-linked ubiquitination. These new data are described and discussed in the Results and Discussion sections.

Figure I: MCC950 promotes K48 linked ubiquitination of NLRP3. Immunoblot analysis of NLRP3, K48-ubq and K63-Ubq in cell lysates (Input) and immunoprecipitated (IP) NLRP3 samples from WT and *Peli2*^{-/-} BMDMs treated with 1 μM of MCC950 for the indicated times.

5. In fig. 8b, a second band appears in the anti-FLAG (Pellino2) blot only in the presence of IRAK1. How do the authors explain this band? Is it phosphorylated Pellino2 (does the band disappear with phosphatase treatment)? Is this band also seen if IRAK1-KD is used instead of wild type IRAK1? Is there a functional relevance of this potential modification?

Response: To directly address the reviewer's comments we show below that the slower migrating form of Pellino2 is not detected when co-expressed with kinase dead IRAK-1 (Figure II). Furthermore, the levels of modified Pellino2, when co-expressed with active IRAK1, are decreased when cell lysates are subjected to phosphatase treatment. Together these data suggest that the second band likely represents a phosphorylated form(s) of Pellino2. This is fully consistent with a number of previous studies that have demonstrated IRAK1 to be capable of phosphorylating Pellino proteins, including Pellino2, on multiple sites (*Strelow et al (2003) FEBS Lett 547,157-61*; *Ordureau et al (2008) Biochem J. 409,43-52*; *Smith et al (2009) PNAS*

106, 4584-90). However, such modification of Pellino2 by IRAK1 is unlikely to be of functional relevance for LPS priming of NLRP3 in the present study since a previous report has already shown that the IRAK1 is not employed by the LPS/TLR4 pathway to phosphorylate Pellino proteins (Goh *et al* (2012) *Biochem. J.* 441,339-46).

Figure II: IRAK-1 promotes phosphorylation of Pellino2. Immunoblot analysis of FLAG, IRAK1 and β -actin in lysates from HEK293T cells transfected with FLAG-tagged Pellino2 and untagged IRAK1 or kinase dead IRAK (IRAK1-KD). Some samples were treated with Lambda phosphatase (λ pp) at 37°C for 30 min prior to immunoblotting.

6. In the same panel, a band in the IRAK1 blot (input) disappears only when wild type Pellino2 is co-expressed. How do the authors explain these findings? Potentially, these results could give further insights into the somewhat unclear mechanism of the Pellino2 / IRAK1 /Nlrp3 interaction.

Response: Since IRAK1 is strongly ubiquitinated by Pellino2, the disappearance of the IRAK band is consistent with slower migration of differentially ubiquitinated forms of IRAK of varying sizes. This manifests as a streaking effect in the immunoblots and we now present the uncropped versions of the input and IP IRAK blots to demonstrate more clearly the Pellino2-induced streaking of IRAK1 and hence reduced levels of bands with discrete mobilities (see **Figure 8b**)

7. The authors propose a model according to which Pellino2 ubiquitinates IRAK1, thus favoring IRAK1 dissociation from NLRP3 (last paragraph of the results section). However, neither do they show that IRAK1 is a substrate of Pellino2 nor that ubiquitination of IRAK1 is indeed the cause for less interaction of IRAK1 with NLRP3. The relevant section(s) should be rephrased to prevent conclusions that are not supported by the data.

Response: As recommended by the Reviewer, this text has now been edited to more accurately reflect the supporting data. Furthermore it should be highlighted that a previous study has demonstrated that IRAK1 is ubiquitinated by Pellino2 (Ordureau

et al (2008) Biochem J. 409,43-52).

8. How do the authors explain the results of the IRAK1 blot (input) in fig. 8g? Was IRAK1-KD expressed?

Response: We welcome the opportunity to clarify this query. This blot was obtained by immunoblotting with an anti-IRAK1 antibody that detects endogenous and over-expressed forms of IRAK1. The first lane shows a single band corresponding to immunoreactivity with endogenous IRAK1 that migrates with a molecular weight of 80 kDa (see **Figure 8h**). It is well known that this represents the unphosphorylated inactive form of IRAK1. The second lane represents samples from cells in which IRAK1 was overexpressed. We and others have previously shown that over-expressed IRAK1 migrates as a slow migrating form of 100 kDa due to hyper/auto phosphorylation. The third lane represents samples from cells in which kinase dead IRAK1 (IRAK1-KD) was expressed and given that IRAK1-KD is not subject to hyperphosphorylation, its expression manifests as increased immunoreactivity at mobility corresponding to 80 kD, the same size as endogenous inactive IRAK1 (see **Figure 8h** for increased levels of 80kDa IRAK1 band in lane 3 versus lane 1).

9. A schematic diagram of the proposed model would enhance readability of the manuscript.

Response: We appreciate this excellent suggestion and an explanatory schematic diagram is now included as Figure 9 in the revised manuscript.

Minor concerns:

a. In several immunoblots it is not clear which bands are the bands of interest (e.g. bands in the IRAK1 blots in fig 8d, NLRP3 blots in fig. 7e and f, etc.)

Response: MW markers are now included in all blots. Furthermore, where required, arrows are now indicated in blots to specify bands/proteins of interest.

b. The overall quality of some of the immunoblots shown is in my opinion suboptimal. For example, the ubiquitin blot in Fig. 7b is overexposed, the signal of the caspase-1 p20 blot in fig 8f is very weak, protein transfer for the ubiquitin blot in fig. 8c seems uneven.

Response: In response to the Reviewer's comments we now include new data to replace Figures 7b and 8f. The β -actin blot in Figure 8c confirms even loading and transfer of samples.

c. The graphs in some figure panels are not correctly or insufficiently labeled (e.g. figs. 4b, 4d, etc.).

Response: We apologise for these oversights. All figure panels are now fully and

correctly labelled.

d. Font size and formatting of the figures are inconsistent, which makes reading more difficult.

Response: All figures are formatted in a consistent style in the revised manuscript.

e. Names of genes or proteins should be used consistently throughout the manuscript (e.g. caspase 1 vs. caspase1 vs. caspase-1).

Response: All genes and proteins are now named consistently in the revised manuscript.

f. Readability would be improved by briefly explaining the RING and FHA mutants in the results section in the context of figure 5.

Response: We now include additional text in the Results section to indicate that the FHA mutant form of Pellino2 lacks substrate-binding activity whilst the RING mutant form lacks E3 ubiquitin ligase activity.

Reviewer #2:

We welcome the reviewer's comments on the originality and importance of our study "*The manuscript describes a novel pathway that regulates NLRP3 activation. The work is of interest to the innate immunity community and addresses an important signaling pathway.*". We are also happy to address the specific points raised by the reviewer.

Major:

Figure 5: The Western blot analysis shows only minor differences in the amount of Caspase p20 between the LPS+ATP stimulated conditions. Especially there seems to be no difference in the amount of p20 between the WT and Peli2-KO BMDMs.

Response: To address these concerns we have now included immunoblots from another replicate of this experiment in Fig. 5c. The data clearly show reduced LPS/ATP-induced processing of pro-caspase-1 to its mature p20 form in Pellino2-deficient cells with this deficiency in processing being rescued by reconstitution of Pellino2-deficient cells with wild type Pellino2 but not with forms of Pellino2 containing mutations in its RING-like or FHA domain. Furthermore, Pellino2-dependent processing of pro-caspase-1 to its mature active form is confirmed by additional data showing reduced processing of pro-IL-1 β into its mature p17 form in Pellino2-deficient cells (**Figures 5c**). Finally, the reproducibility of these studies is now demonstrated by new data (**Figure 5d**) that represent densitometric analysis of caspase-1 p20 across all replicates of this experiment.

Figure 7(a,b,d,e):

Show size markers for Western blots and check whether the ubiquitin signal only appears at a higher molecular weight than NLRP3. If ubiquitin signal is detected at a lower molecular weight than NLRP3 this indicates that NLRP3 co-immunoprecipitates with other ubiquitinated proteins and a definite answer about the ubiquitination of NLRP3 is not possible.

Please provide uncropped images of NLRP3 to make sure that ubiquitinated NLRP3 is detected at a higher molecular weight compared to unmodified NLRP3.

As an alternative, the authors could perform an IP for ubiquitin itself and stain the Western-blot for NLRP3 (see Emmerich and Cohen, 2015).

Response: Firstly, molecular weight markers are now included for all Western blots and it is clear that most of the ubiquitination signal that is induced by LPS manifests at sizes above the molecular weight of NLRP3. Furthermore, we wish to emphasise that prior to immunoprecipitating NLRP3 for assay of ubiquitination status, we treat extracts with 1% (w/v) SDS and heat to 95°C for 5 min to dissociate NLRP3 from any interacting proteins. Lysates are then diluted in lysis buffer before immunoprecipitation and western blotting for ubiquitin (described in “*Immunoblotting and Immunoprecipitation*” subsection of *Methods* Section). This ensures that the ubiquitin signal represents covalent attachment of ubiquitin molecules to NLRP3 and not associated proteins. The efficiency of this protocol in removing associated proteins is demonstrated below in which LPS is shown to promote the interaction of NLRP3 with IRAK1 (by their co-immunoprecipitation) but this interaction is no longer observed when extracts are treated with SDS prior to immunoprecipitation (Figure III).

Figure III: SDS pre-treatment precludes co-immunoprecipitation of interacting proteins. Immunoblot analysis of NLRP3 and IRAK1 in lysates (Input) and IP IRAK1 samples from WT BMDMs treated with 100ng/ml LPS for the indicated times. Prior to immunoprecipitation, samples were treated with or without 1% (w/v) SDS followed by heating to 95°C for 5 min

As requested by the Reviewer, uncropped images of NLRP3 are now included and these images show that most of the ubiquitin signal induced by LPS is apparent at sizes above unmodified NLRP3. There is some modest ubiquitination corresponding to sizes below NLRP3 but given that we have treated samples with SDS to dissociate NLRP3-interacting proteins, such signal at lower molecular weights likely represents degraded / processed forms of ubiquitinated NLRP3.

To further confirm the specific ubiquitination of NLRP3, we performed the very valuable studies recommended by the Reviewer in which ubiquitin chains were precipitated and Western blotted for NLRP3. We also exploited this approach to define the type of linkages in the polyubiquitin chains that are attached to NLRP3. Thus, using K63-specific TUBEs we isolated K63-linked ubiquitinated proteins followed by immunoblotting for NLRP3 and now include new data to show that LPS promotes time dependent K63-linked ubiquitination of NLRP3 in wild type macrophages and this is reduced in Pellino2-deficient cells (see new **Figure 7b**). Similar approaches using K48-specific TUBEs failed to detect any K48-linked ubiquitination of NLRP3 in response to LPS (see new **Figure 7c**) demonstrating that LPS promotes K63-linked ubiquitination of NLRP3 in a Pellino2-dependent manner. As described in Response 4 to Reviewer #1, similar TUBE-based approaches are also now included to show that MCC950 strongly suppresses the ability of LPS to induce the K63-linked ubiquitination of NLRP3 (see new **Figure 7f**).

Figure 7:

The authors should check the ubiquitin linkage type resulting from Pellino2. Please use linkage type specific antibodies or TUBEs to check for K48 and K63 ubiquitination. The activating mechanism of ubiquitination would be more convincing in case NLRP3 would be modified with an atypical linkage type, as a K48 ubiquitination should only result in enhanced degradation of NLRP3.

Response: We appreciate this excellent suggestion from the Reviewer and as described above we now include new data using K48- and K63-specific TUBEs to show that LPS promotes K63-linked ubiquitination of NLRP3 in a Pellino2-dependent manner (see new **Figures 7b, 7c, 7f**). We also show that MCC950 strongly suppresses the ability of LPS to induce the K63-linked ubiquitination of NLRP3 (see new **Figure 7f**).

Figure 8e:

Show TNF α secretion as a priming control for the experiment.

Response: We now include TNF α as a priming control for this experiment (see **new Figure 8f, lower panel**). Whereas IRAK1 deficiency results in enhanced levels of IL-1 β in response to LPS/ATP (upper panel Figure 8f) the absence of IRAK1 leads to reduced levels of TNF α (lower panel Figure 8f). This is in keeping with the receptor

proximal role of IRAK1 in the TLR4 pathway that drives induction of TNF α and is consistent with previous reports in the field. (see Kalantari *et al* (2017) J. Biol. Chem 292(14):5634-5644). It should also be noted that on foot of a recommendation from Reviewer #3, these new data were obtained using primary IRAK1-deficient BMDMs and replace the data in the original submission that were generated from immortalized *Irak1*^{-/-} BMDMs.

Figure 8(b,c,g,h):

Show size markers for Western blots and check whether the ubiquitin signal only appears at a higher molecular weight as the precipitated proteins (NLRP3/IRAK1). If ubiquitin signal is detected at a lower molecular size than the target protein, this indicates that NLRP3/IRAK1 co-IP with other ubiquitinated proteins and a definite answer about the ubiquitination of these proteins is not possible.

Response: Molecular weight markers are now included for all Western blots and confirm that the ubiquitin signal is largely above the size of the target protein. Furthermore, as described above, we treated cell extracts with 1% (w/v) SDS and heated to 95°C for 5 min prior to immunoprecipitation of the target protein in order to preclude any interacting proteins contributing to the ubiquitin signal. Finally, new data are now included using TUBEs to further confirm specific ubiquitination of NLRP3 (**Figures 7b, 7c, 7f and 7g**) and IRAK-1 (**Figure 8d**)

Minor:

Page 4:

The authors state that there is no understanding of E3 ligases which ubiquitinate NLRP3. However, previous studies identified already TRIM31 (Song, 2016, Nat. Comm.), FXBL2 (Han et al., 2015, JBC) and March7 (Yan et al., 2015, Cell) as NLRP3 E3 ligases regulating NLRP3. Please include these Papers in your discussion.

Response: These papers are now cited and described in the *Introduction* and *Discussion* of the revised manuscript.

Page 5/ Figure 1f:

The authors state that the levels of mature IL-1beta were strongly reduced in the Pellino2-KO. The secreted IL-1beta might be as well pro-IL1beta from dying cells. Especially since the amount of secreted IL-1beta is not altered upon LPS stimulation in the rest of the study.

Response: Text has been revised to specify reduced levels of “secreted” IL-1 β in Pellino-deficient cells. It should also be noted that Figure 1f represents the levels of secreted IL-1 β after stimulation of cells with LPS for 24h. This time exposure results in production of IL-1 β in response to LPS alone. In contrast, the remaining studies used the 2 signal LPS/ATP model with cell exposure to LPS being limited to 3 hours. Such acute exposure to LPS alone does not generate secreted IL-1 β and requires an

additional second signal like ATP. This provides an explanation for the production of IL-1 β in response to LPS alone in Figure 1f but not in subsequent studies with 3h stimulation by LPS.

Page 10:

The authors state that NLRP3 might be a direct target of Pellino3. This is supposed to be Pellino2.

Response: This correction has been included in the revised manuscript.

Figure 4b:

The figure is not labeled.

Response: Figure 4b is now fully labelled in the revised manuscript

Figure 8b:

Label the figure with “IP IRAK1” and not just “IP IRAK”

Response: This correction has been included in the revised manuscript.

Reviewer #3:

We welcome the reviewer’s comments on “the findings are certainly novel” and are happy to address the specific points raised by this reviewer.

1. Without any molecular weight markers, it is difficult to ascertain the accuracy of the Western blots presented in all figures. Please add these to all blots.

2. Response: Molecular weight markers are now included for all Western blots.

Please clarify in the figure legends on how many separate occasions the *in vivo* experimental results were verified (e.g. Figure 3, figure 4). Why are there different numbers of mice (data points) for each cytokine measurement in Figures 3 and 4 (presumably all cytokines were assessed for each mouse so there should be an equal number of dots for each analysis)?

Response: Details are now included in relevant legends to indicate that *in vivo* data in Figure 3 represent pooled biological replicates from 2 independent experiments and the *in vivo* data from Figure 4 is representative of 2 independent experiments (with the replicate experiment included below (Figure IV)).

Figure IV: ELISA of IL-1 β , IL-18 and IL-6 in peritoneal lavage from WT and *Peli2*^{-/-} mice previously infected for 10h by intraperitoneal injection of PAO1 (1.5 x 10⁷ CFU)

There is some variation in numbers of mice for each cytokine measurement since there was some variation in the volume of serum recovered from each mouse and so this precluded the assay of all of the indicated cytokines in some serum samples of reduced volume.

3. Figure 1e. Why doesn't LPS result in IFN β production?

Response: Poly (I:C) is a very strong inducer of IFN- β production in BMDMs with other TLR ligands being relatively ineffective. LPS induced modest levels of IFN- β but is dwarfed by Poly (I:C). In order to clearly demonstrate the ability of LPS to induce IFN- β , Figure 1e now illustrates in separate panels (allowing for different y-axes scales) the induction of IFN- β in response to Poly (I:C) and LPS. Pellino2 deficiency has no effect of the ability of either ligand to induce IFN- β .

4. Figure 1f. Wildtype BMDMs should not respond to LPS to cause IL-1 β secretion (i.e. LPS primes/induces precursor IL-1 β production, but a separate inflammasome activator, signal 2, is normally required for its activation and secretion). Are the authors using ultrapure LPS or LPS that is contaminated with other bacterial ligands/PAMPs? Why do some assays show robust IL-1 secretion with LPS alone (e.g. figure 1 [over 2000-6000pg/ml], figure 2k, figure 7c LPS with MCC930 treatment [very strange]), while others show none (e.g. Figure 2a/e, supp fig 4 etc)?

Response: It should be noted that Figure 1f represents the levels of secreted IL-1 β after stimulation of cells with LPS for 24h (new Figure 1f now showing pooled data from 3 independent experiments to confirm robustness and reproducibility). This time exposure results in production of IL-1 β in response to LPS alone. In contrast, many of the remaining studies used the 2-signal LPS/ATP model with cell exposure to LPS being limited to 3 hours. Such acute exposure to LPS alone does not generate secreted

IL-1 β (e.g. Figure 2a, e, i, Supp. Fig. 4d) and requires an additional second signal such as ATP to produce IL-1 β (e.g. Figure 2a, 7c). Overnight stimulation with LPS versus more acute LPS/ATP co-stimulation are different experimental systems making it difficult for direct comparison of absolute levels of IL-1 β . Furthermore, the Reviewer also cites Figure 2k. It should be noted that these data refer to the human monocytic cell line THP1 making it challenging for meaningful comparison of levels of IL-1 β across human cell lines and primary murine BMDMs. Finally, all studies utilize ultrapure LPS from Enzo.

5. Page 6. The authors write “Given that Pellino2 is required for LPS induced production of mature IL-1 β but is redundant for the initial expression of pro-IL-1 β ”. No data is provided prior to this statement that shows Pellino2 being required for IL-1b maturation (i.e. Western blots showing cleavage to the active p17 fragment, or lack thereof in Pellino2 KOs).

Response: We welcome the opportunity to correct this oversight. Since the data focusing on the maturation of IL-1b comes later in the manuscript the text has now been changed to read “*Given that Pellino2 is not required for induction of pro-IL-1 β we next probed the role of Pellino2 in the activation of inflammasomes.*”

6. Figure 2. The error bars (for figure 2 ELISAs where experimental triplicates are shown) are meaningless as they just demonstrate that the authors know how to pipette. The authors should show true biological replicates or pool the independent experiments and, in either case, present each experiment/mouse as a dot point – this would allow readers to accurately assess the true biological variability within experiments and between genotypes. I strongly recommend that this methodology be applied to all figures, particularly where technical repeats are currently shown.

Response: As recommended by the Reviewer, biological replicates are now included for all experiments with each experiment/mouse being represented by a dot.

7. Figure 2d/h. In my opinion these are the critical experiments as they detect mature IL-1 and caspase-1 in WT and Pell2 KO cells. However, the Pell2 KO lanes containing the pertinent treatments (nigericin/atp) appear under-loaded relative to the similarly treated WT controls (i.e. cell lysate actin, caspase-1 and pro-IL-1b levels are all reduced compared to WT cells). Therefore it is not surprising less cleaved caspase-1 and IL-1 are detected in the supernatants in the Pell2 KO cells. Could the authors please provide better quality Westerns from independent experiments where the loading does not cloud the interpretation of the data. Detailed ATP/Nigericin dose curves and kinetics of IL-1 and caspase-1 maturation/secretion between WT and Pell2 KO cells would be helpful data that would strengthen the authors findings and conclusions. A particulate NLRP3 activator (such as Alum) should also be tested to define how important Pell2 is for both soluble and particulate NLRP3 stimuli.

Response: As recommended by the Reviewer Figure 2d and Figure 2h now show improved quality western blots with consistent loading controls across all samples.

It should be noted that the dose and time treatments for ATP and Nigericin are consistent with conditions that are widely used in the literature. Furthermore, we confirm below, with dose response curves, that the concentrations of ATP and Nigericin used in the manuscript are optimal for IL-1 release and that the reduced levels of IL-1 β in Pellino2-deficient cells are observed across various concentrations of ATP and Nigericin (Figure V).

Figure V: ELISA of IL-1 β of medium from WT and *Pelii2*^{-/-} BMDMs treated with 100ng/ml LPS for 3 h with or without further stimulation with ATP or Nigericin at the indicated concentrations for 1 h.

New data is now included to show that alum (NLRP3 particulate stimulus) also depends on Pellino2 in order to mediate its maximal secretion of IL-1 β (see new **Figure 2i**).

8. Figure 2k/l. The results are confusing. Figure 2k shows LPS alone induces secretion of IL-1 (which is shouldn't) and that the addition of ATP or Nigericin treatment has no effect (which it should), while in contrast, figure 2l shows that only LPS/ATP (not LPS alone) cause maturation and secretion of IL-1 and caspase-1? Figure 2l contains ATP and nigericin treatments, while figure 2l only shows ATP treatments? Please clarify.

Response: We wish to highlight that the data in Figure 2k/l (now Figures 2l and 2m) were obtained from the THP1 human monocytic cell line. Gaidt et al (2016; *Immunity* 44:833) have previously shown that in response to LPS, human monocytes can secrete IL-1 β independently of classical inflammasome stimuli. We now include pooled data from 3 experiments confirming the ability of LPS alone to be capable of secreting IL-1 β in THP1 monocytes (see new **Figure 2l**). Co-stimulation of cells with LPS and ATP or nigericin slightly enhanced the levels of IL-1 β but this was reduced in Pellino2-deficient cells. Interestingly, unlike the ELISA analysis (**Figure 2l**), we were unable to detect processed IL-1 β and caspase1 by Western blotting when THP1 cells were stimulated with LPS alone (see new **Figure 2m**) but this may reflect the

greater sensitivity of the ELISA system. Co-stimulation of cells with LPS and ATP or nigericin was sufficient to produce detectable levels of processed IL-1 β and caspase1.

9. Supp. Figure 4. Transfected LPS activates caspase-11, which in turn activates NLRP3-caspase-1 to cause IL-1 maturation and secretion. Wouldn't the authors' model predict that, contrary to results presented, transfected LPS (which specifically activates NLRP3) should result in reduced IL-1 secretion in Pell2 KO cells? The *in vivo* data suggests this should be the case (Figure 3) as LPS lethality is largely dependent on caspase-11.

Response: The Reviewer raises an interesting question. Whilst caspase-11 induces some direct effects such as Gasdermin D cleavage and pyroptosis, it has also been proposed to promote NLRP3-dependent processing of caspase1 and pro-IL-1 β and as the Reviewer highlights this may lead one to predict that Pellino2 deficiency would result in reduced IL-1 β secretion in response to transfected LPS. However, the mechanism by which caspase-11 activates the NLRP3 pathway remains to be fully delineated and the lack of effect of Pellino2 deficiency on this non-canonical pathway may reflect a different activating mechanism than the Pellino2-dependent mechanism employed in the canonical NLRP3 pathway that is triggered by LPS/ATP. Furthermore, Pellino2 deficiency did not fully protect against LPS induced lethality *in vivo* indicating that the enhanced survival observed in Pellino2-deficient mice may be primarily attributed to reduced IL-1 β production *in vivo* rather than effects on pyroptosis and IL-1 α release. Indeed, IL-1 α production and pyroptosis are independent of NLRP3. This discussion is now included in the revised manuscript.

10. Figure 4B. Western blot needs labeling.

Response: Figure 4b is fully labelled in the revised manuscript

11. Figure 4f. Pell2 proIL1 is reduced, suggesting this is the reason for reduced cleavage. The caspase-1 p20 blot is of very poor quality.

Response: An independent replicate of this experiment showing comparable levels of pro-IL-1 β and a higher quality caspase-1 p20 blot is now included in Figure 4f.

12. Figure 5C. The caspase-1 p20 fragment is running at different sizes (e.g. WT vs Pell2 KO controls) and differences in cleavage (e.g. band intensity) are not convincing. Please show the other two independent experiments to help convince readers of differences observed. Blots examining IL-1 cleavage and secretion into the supernatant are lacking, which is strange as the authors have blotted lysates for pro-IL-1 in this experiment.

Response: To address these concerns we have now included immunoblots from another replicate of this experiment in Fig. 5c. The data clearly show reduced LPS/ATP-induced processing of pro-caspase-1 to its mature p20 form in Pellino2-deficient cells with this deficiency in processing being rescued by reconstitution of Pellino2-deficient cells with wild type Pellino2 but not with forms of Pellino2 containing mutations in its RING-like or FHA domain. Furthermore,

Pellino2-dependent processing of pro-caspase-1 to its mature active form is confirmed by additional data showing reduced processing of pro-IL-1 β into its mature p17 form in Pellino2-deficient cells (**Figures 5c**). Finally, the reproducibility of these studies is now demonstrated by new data (**Figure 5d**) that represent densitometric analysis of caspase-1 p20 across all replicates of this experiment.

13. Figure 6D. Levels of soluble ASC are lower in Pell2 KO – which makes the data difficult to interpret (i.e. loading appears less than WT cells). The ASC blot (soluble) is also overexposed which probably means the loading differences are even greater than what is already apparent.

Response: To address this concern and also demonstrate the reproducibility of the findings we now include an independent replicate of this experiment (see **new Figure 6D**) that shows comparable levels of ASC in WT and Pellino2-deficient cells with images being moderately exposed. It should be noted that activation of NLRP3 by LPS/ATP leads to re-distribution of ASC into an insoluble fraction and this is reflected by lower levels of soluble ASC under these conditions. These data clearly show Pellino2-mediated ASC oligomerization that is dependent on its FHA and RING-like domains.

14. Figure 7a/b/d/e. This data is difficult to interpret without molecular weight markers to indicate the sizes of the ubiquitin laddering. Figure 7a is very convincing whereas figure 7b is less so (i.e. in figure 7a there appears to be a complete lack of NLRP3 ubiquitination in Pell2 KO cells after 3 hrs of LPS treatment, whereas in 7b after 3 hrs of LPS treatment the ubiquitination of NLRP3 in Pell2 KO cells, although perhaps somewhat reduced, is still quite apparent). Which data set is representative of Pell2 KO cells? I would like to see several independent experiments to gauge the significance/reproducibility of this data.

Response: Firstly, molecular weight markers are now indicated on all immunoblots. In order to address the comments of the reviewer we have also carefully examined a number of replicate experiments (in which NLRP3 is initially immunoprecipitated followed by immunoblotting for ubiquitin) as well as performing complementary studies using TUBEs to isolate ubiquitinated proteins followed by immunoblotting by NLRP3. Across these studies, the most consistent time at which we detect maximal ubiquitination of NLRP3 in response to LPS is 90 min with this ubiquitination being impaired in Pellino2-deficient cells. We show clear Pellino2-dependent ubiquitination of immunoprecipitated NLRP3 at 90 min post LPS stimulation (see **new Figure 7a**). We also exploited TUBEs to define the linkages in the polyubiquitin chains that are attached to NLRP3. Using K63-specific TUBEs we isolated K63-linked ubiquitinated proteins followed by immunoblotting for NLRP3 and now include new data to show K63-linked ubiquitination of NLRP3 at 90 min post LPS stimulation in wild type macrophages and this is reduced in Pellino2-deficient cells (see **new Figure 7b**). Similar approaches using K48-specific TUBEs failed to detect any K48-linked ubiquitination of NLRP3 in response to LPS (see **new Figure 7c**) demonstrating that LPS promotes K63-linked ubiquitination of NLRP3 in a Pellino2-dependent manner.

Similar approaches were used to study the regulatory effects of MCC950 (see **Figure 7e** and **new Figure 7f**) and again LPS was shown to promote strong ubiquitination of NLRP3 at 90 min confirming the significance and reproducibility of the data.

15. Figure 7d/e. Why does MCC950 alone result in strong NLRP3 ubiquitination (comparable to control LPS treated cells) and what does this mean given that in this scenario LPS treatment in the presence of MCC950 appears to actively trigger the de-ubiquitination of NLRP3 (particularly as MCC950 potentially inhibits NLRP3 activation)? Why does NLRP3 ubiquitination disappear after 3 hrs of LPS treatment in the control treated cells (in contrast to Figure 7a/b), which is often when the NLRP3 activating stimulus (e.g. ATP) is added to cells which then reportedly triggers NLRP3 deubiquitylation?

Response: As described above and indicated by this reviewer, MCC950 inhibits the LPS-induced ubiquitination of NLRP3 whilst also promoting some basal ubiquitination of NLRP3 in unstimulated cells (see **Figure 7e**). However, we now include new data from the use of K63-specific TUBEs to isolate K63-linked ubiquitinated proteins to show that LPS promotes strong and time dependent-ubiquitination of NLRP3 and this is fully abrogated in cells that are pre-treated with MCC950 (see **new Figure 7f**). Notably, MCC950 alone failed to induce any K63-linked ubiquitination of NLRP3 suggesting that the basal ubiquitination of NLRP3 observed in response to MCC950 is due to ubiquitin chain with different linkages. Indeed, as described above, we include new data showing that MCC950 alone induces time-dependent K48-linked ubiquitination of NLRP3 (see Figure I above in response to Reviewer#1, point 4). These data are consistent with a model in which LPS promotes transient K63-linked ubiquitination of NLRP3 that is associated with NLRP3 activation and this is countered by the inhibitory effects of MCC950 that opposes the K63-linked ubiquitination of NLRP3 and instead biases towards its K48-linked ubiquitination. These new data are described and discussed in the Results and Discussion sections.

16. Figure 7f. Is it expected that Pell2 associates with NLRP3 after 0.5hr of LPS treatment and that this association is then lost after 2 hrs of LPS treatment, as the data suggests (particularly as NLRP3 ubiquitination, according to Figure 7b, is only strongly detected at 3 hrs [although the data in 7e suggests it is ubiquitinated within 1 hr and disappears at 3 h – which result is representative?]? How robust are these results (please show the additional 2 experimental repeats of this data; 7f, 7b, 7e)?

Response: As indicated above our various studies suggest that ubiquitination of NLRP3 in response to LPS is at its maximum at 90 min. We have thus performed more detailed time course experiments to characterize the binding of Pellino2 to NLRP3 over this time frame (see **new Figure 7g**). We now include new data to show that LPS induces time-dependent interaction of Pellino2 over these times with strong binding being still evident at 90 min post stimulation that is consistent with the earlier described ubiquitination of NLRP3 at this time. In addition, we also include important

new data using the complementary approach of TUBE analysis to further characterize K63-linked ubiquitination of NLRP3 at 90 min post LPS stimulation (see **new Figures 7b and 7f** for reproducibility)

17. Figure 8c. From this data, the authors conclude that “the greatly reduced ubiquitination of IRAK1 in *Peli2*^{-/-} cells still permitted degradation of IRAK1, albeit with slightly delayed kinetics and more residual IRAK at later time points than in WT cells.” I am not convinced that the data shows this. If anything, to my eye there appears to be more residual IRAK1 in WT cells (although levels are almost undetectable). Please provide the independent experiments of this data to demonstrate the robustness of this data and the conclusions that have been drawn from it.

Response: We present below 3 replicates (1-3) of the effects of Pellino2 deficiency on LPS-induced degradation of IRAK1 and also include pooled densitometric data (bottom panel) from the 3 experiments (Figure VI).

Figure VI: Effects of Pellino2 deficiency on LPS induced degradation of IRAK1. Immunoblot analysis of IRAK1 and β-actin in lysates from WT and *Peli2*^{-/-} BMDMs treated with 100ng/ml LPS for the indicated times. 3 independent studies are shown with pooled densitometric analysis of normalized IRAK1 intensity being shown in the bottom panel.

Although the differences are modest the data suggest that the basal levels of IRAK1 and residual levels post LPS (especially at early times and up to 90 min) are consistently higher in Pellino2-deficient cells. We also present new data using TUBEs to complement these approaches and further confirm the contribution of Pellino2 to LPS-induced ubiquitination of IRAK1 (see **new Figure 8d**).

18. Figure 8d. How can IRAK1 associate with NLRP3 after one, two and three hours of LPS stimulation when it is almost completely degraded within 15 minutes of LPS stimulation?

Response: Whilst stimulation of cells with LPS results in greatly reduced levels of IRAK1, some residual IRAK1 is detectable post stimulation. It should also be noted that IRAK is subject to multiple forms of post-translational modification, especially phosphorylation and ubiquitination that will cause electrophoretic mobility shifts that can be distributed across slower migration regions of the blot. Thus, these modified forms will not be visible at the region corresponding to native IRAK1 but will still be subject to immunoprecipitation and detection of interacting proteins such as NLRP3.

19. Figure 8e/f. These experiments are performed with immortalized BMDMs which have been known to exhibit variable levels of ASC. Please blot and compare levels of total ASC and caspase-1 in WT, IRAK1 and IRAK4 iBMDMs (control blots that are currently lacking if 8f). This is important as previous work using primary BMDMs from IRAK1 deficient mice have reported that IRAK1 is required for efficient NLRP3 responses, including ASC oligomerisation (e.g. PMID: 24043892). The authors should examine primary macrophages derived from IRAK1 deficient mice to ensure their results using iBMDMs are not just a consequence of immortalization and/or genetic drift.

Response: In response to the Reviewers comments and to avoid any possibility of misleading conclusions being derived from immortalized BMDMs, we have now performed the studies requested by the reviewer; in primary BMDMs from IRAK1-deficient mice (see **new Figures 8f and 8g**). **Figure 8f** confirms augmented secretion of IL-1 β in IRAK1-deficient BMDMs in response to LPS/ATP or LPS/nigericin with **Figure 8g** demonstrating that augmented secretion is associated with greater processing of IL-1 and caspase-1. These data fully support the original conclusions of our study.

It should be noted that in the report cited above by the Reviewer, the analysis of NLRP3 responses in IRAK deficient cells was limited to rapid caspase 1 cleavage in response to acute co-stimulation with LPS+ATP with no data being provided on levels of secreted IL-1 β . Furthermore, the same study demonstrated that stimulation with LPS for 3 hours followed by ATP resulted in enhanced caspase-1 cleavage in IRAK1-deficient BMDMs compared with WT cells. The latter data are consistent with our observations.

20. Figure 8h. Why isn't NLRP3 ubiquitylated in response to LPS treatment in WT cells (in contrast to the data provided in figure 7). Again, how do the authors explain robust ubiquitylation at 1.5h (this time in IRAK1 KO cells) and then a loss of this signal at 3 hr (which is normally when the NLRP3 signal 2 is provided)?

Response: A more detailed kinetic study is now included (**Figure 8i**) confirming that LPS induces ubiquitination of NLRP3, especially at 90 min in WT cells and this is further enhanced in IRAK-1 deficient cells. This timeframe is consistent with the kinetics of NLRP3 ubiquitination in various data panels in Figure 7. We propose that Pellino2-dependent K63-linked ubiquitination of NLRP3 by LPS is a transient but important part of the priming phase of NLRP3 activation, possibly by limiting the binding and negative regulatory effects of IRAK for a period of time during the priming process. The role of Pellino2-mediated ubiquitination of NLRP3 may possibly extend to recruiting other proteins such as ASC and Caspase 1, that both interact with NLRP3 during the priming phase. We propose that, upon stimulation with LPS, NLRP3 is K63 ubiquitinated followed by deubiquitination in preparation for inflammasome assembly.

21. Figure 8i. Please provide the independent experiments and a proper time course of NLRP3 binding to Pell2 (not just the 30 min LPS treatment shown) to define its association with NLRP3 when NLRP3 ubiquitylation is observed.

Response: As recommended by the Reviewer, we now include new data of a detailed time course demonstrating time dependent interaction of Pellino2 with NLRP3 in response to LPS in WT cells (see **new Figure 8j**). The interactions are further enhanced in IRAK1-deficient cells. Furthermore the interactions occur at times (30-90 mins) that are consistent with times of LPS-induced ubiquitination of NLRP3 (see Figures 7a, 7b, 7e, 7f).

In summary, we welcome the reviewer's very constructive comments and suggestions. We feel that the inclusion of a considerable amount of new data addresses all of their concerns in a comprehensive manner, adds further support to our original conclusions and greatly improves the quality and novelty of the manuscript. We hope you will consider the revised manuscript worthy of publication in *Nature Communications*.

Kind Regards

Prof. Paul Moynagh

Corresponding Author

Reviewers' comments:

Reviewer #1 (Remarks to the Author):

The authors have carefully revised their paper in response to the reviewers comments. However, some points are still open and addressing those would significantly improve the paper. I refer to my previous comments and provide my evaluation according to those:

Major concerns:

1. OK

2. While the cytokine data in figure 5b and the IL-1b processing data in figure 5 are convincing, the data on caspase-1 cleavage in figure 5c in conjunction with the quantification of three replicate experiments in figure 5d are not really convincing. Even though figure 5c now shows immunoblots from an experiment where differences are evident, figure 5d indicates, that differences in caspase-1 cleavage are clear only in one of three experiments. For a fair comparison, the extent of caspase-1 cleavage should be statistically analyzed and compared between WT and Peli2-KO cells and not between Peli2-KO cells and reconstituted Peli2-KO cells, because in reconstituted a Peli2-KO cells caspase-1 cleavage is higher than in WT cells (possibly an artifact of overexpression).

Given the data from three experiments, the text does not appropriately represent the experimental data with respect to caspase-1 processing ("This is further supported by the observation that the re-introduction of WT Pellino2 could re-constitute the LPS/ATP-induced processing of pro-caspase-1 in Peli2-/- BMDMs to the same degree as displayed in WT cells whereas both the FHA and RING-like mutants of Pellino2 were ineffective in this regard (Fig. 5c, 5d).").

The data on IL1b-p17 from reconstituted Peli2-KO cells are convincing.

3. See 2. Caspase-1 processing data are still not convincing. IL-1b is OK.

4. While the data on the effects of MCC950 on the linkage-specific ubiquitination of NLRP3 are interesting, I do not see how these results add to the understanding of the function of Pellino2 in inflammasome activation or NLRP3 ubiquitination. Thus, and for conciseness sake, I suggest taking out these data altogether.

5. The study of Ordureau et al. (cited by the authors) shows that in the case of Pellino1, phosphorylation by IRAK4 clearly enhances Pellino1's E3 ligase activity. I still believe that it would add mechanistical depth to test if IRAK1-dependent phosphorylation of Pellino2 is also relevant for Pellino2's E3 activity and subsequent NLRP3 inflammasome activation, independently of whether or not Pellino2 is directly phosphorylated by IRAK1 in this context.

6. OK

7. The changes to the text are OK. The study cited by the authors (Ordureau et al (2008) *Biochem J.* 409,43-52), however, does not show that IRAK1 is ubiquitinated by Pellino2. This would require an in vitro assay with purified components, overexpression in HEK293T cells does not prove a direct enzyme-substrate relationship.

8. OK

9. The schematic in the current form is not intuitively understandable. Especially the use of different styles and colors of arrows is confusing/not explained. What does e.g. a solid black arrow mean? 'Activates', 'recruits', 'ubiquitinates', 'leads to', etc.? Also it would maybe be a good idea to show two panels showing the course of events in WT and Pellino2-KO cells, respectively.

Minor comments:

a. OK

b. The b-actin band migrates around 42kDa and is therefore not a good control to assess transfer of proteins above 70kDa. A ponceau staining of the whole membrane would be more appropriate. Clearly there are issues with the ubiquitin blot in figure 8c (shades/white areas). Not publication quality.

c. Figure 4d still says Capsase 11!

- d. Font sizes in the figures still vary widely (check for example legend of bar graphs vs. immunoblots in figure 2, 4, etc.; font size in fig. 2h vs. fig. 2m; figure 8)
- e. OK
- f. OK

Reviewer #2 (Remarks to the Author):

The authors have appropriately answered all my questions by performing new experiments or providing explanations.

The paper will be received very well by the field.

Reviewer #3 (Remarks to the Author):

The authors have performed a large number of additional experiments in order to address the concerns raised. Although this is commendable, I find it disconcerting how in some cases the new data sets are dramatically different to those presented in the original manuscript (see below). However, even in the revised manuscript there remain data sets where the effect of Pell 2 deficiency on NLRP3 activity is not particularly convincing, or where two data sets show quite different results (e.g. point 5 below).

Comments:

1. Figure 2d/h. Unfortunately, the quality of these new blots is still quite poor. In particular, the supernatant IL-1 (two bands?) and caspase-1 bands are hazy/out of focus, and whether there is a real difference between WT and Pell 2 KO cells difficult to ascertain. Given the results are not black and white in WT and Pell 2 KOs, the gels should also have included an appropriate control (i.e. activation of an inflammasome that isn't impacted by Pell2 deficiency). Similarly, Figure 2m shows little, if any, difference between the caspase-1 and IL-1 cleavage fragments in the supernatants. Clear-cut data on this issue is essential for the authors to conclude that Pell 2 impacts NLRP3 activity (i.e. if clear demonstrable reductions in NLRP3 induced caspase-1 processing are not observed then this calls into question the manuscript's main conclusion). I also find it strange that if NLRP3 activation is impacted by the loss of Pell2, then why do WT and Pell2 KO cells secrete equivalent levels of IL-1 at low doses of nigericin, as their rebuttal data demonstrates?
2. Figure 2l/m. I still fail to see why the western blots don't detect cleaved IL-1/casp1 with LPS stimulation alone given that the levels of IL-1 detected by ELISA are almost on par (and were identical in the original manuscript) with the LPS/ATP, LPS/Nigericin stimulations.
3. Original point 9. Caspase-11 activates NLRP3 via potassium efflux, which is how nigericin and ATP also activate NLRP3. This data actually better fits with the western blots and low dose nigericin treatments suggesting Pell2 loss results in normal levels of NLRP3 activity in several situations.
4. Figure 5C. It is very difficult to know which data set to believe. The original blot showing normal caspase-1 processing and secretion in Peli2 KO cells, or the new blot in the revised manuscript showing undetectable caspase-1 processing and secretion? The contradictory nature of these two data sets (and others, such as the original Figure 7f versus revised 7g) calls into question how consistent some of the results are.

5. Figure 6b-d. I find it concerning that one experiment with LPS/ATP shows a moderate reduction in ASC crosslinking upon Pell2 loss (b) while a second experiment shows a complete loss in ASC crosslinking (c).

6. The authors show Pell2 dependent ubiquitination of NLRP3, and write that "These data are consistent with a model in which LPS promotes transient K63-linked ubiquitination of NLRP3 that is associated with NLRP3 activation.....". I'm not sure this statement is warranted because LPS alone does not activate NLRP3. In addition, the authors data suggest that LPS triggers K63-ubiquitination of NLRP3 in 1.5hrs and that this completely disappears by 3 hrs. Therefore, it is very much a black hole as to what this NLRP3 ubiquitination is actually doing without experiments to define the E3 ligase responsible, or the lysine's modified and what happens to NLRP3 activity if these sites are mutated.

7. Figure 8d. I still find it unclear how IRAK1 can be immunopurified from cells with almost undetectable IRAK1 levels. The authors state that "...IRAK...modified forms will not be visible at the region corresponding to native IRAK1 but will still be subject to immunoprecipitation and detection of interacting proteins such as NLRP3." The IRAK1 doublet detected in the IP and the input blot are running at the same molecular weight (70-80kDa) implying that they are the same form that is almost completely degraded upon LPS treatment. The 100 kDa IRAK1 form that doesn't appear to be degraded (e.g. Figure 8e, 8i) and is therefore arguably more likely to be available for complexing and IP doesn't actually appear to be detected in the IRAK1 IP samples (8e).

22/12/2017

Dear Editor,

Re: NCOMMS-17-12750A

I wish to appeal the decision indicated in your letter of 21st November 2017 on the above manuscript. To assuage all concerns of reviewers, we have now included additional replicate experiments with sufficient statistical Power to confirm that all conclusions are fully supported by statistical significance. We have also optimised some of our sample collection to further enhance the quality of blots of samples of conditioned media from cells and hope these efforts demonstrate our willingness to fully satisfy all concerns of Reviewers. In addition to providing point-by-point responses below and a revised manuscript and figures, for your convenience I have also separately included an “Additional Data” file that includes all new data (together with uncropped blots to provide sense of overall quality of our data).

Responses to the individual points raised by each of the reviewers are now detailed below:

Reviewer #1:

We welcome the reviewer’s comments that “*The authors have carefully revised their paper in response to the reviewers comments..*”. We are also happy to address the specific points raised by the reviewer.

Major concerns:

1. OK

2. While the cytokine data in figure 5b and the IL-1b processing data in figure 5 are convincing, the data on caspase-1 cleavage in figure 5c in conjunction with the quantification of three replicate experiments in figure 5d are not really convincing. Even though figure 5c now shows immunoblots from an experiment where differences are evident, figure 5d indicates, that differences in caspase-1 cleavage are clear only in one of three experiments. For a fair comparison, the extent of caspase-1 cleavage should be statistically analyzed and compared between WT and Peli2-KO cells and not between Peli2-KO cells and reconstituted Peli2-KO cells, because in reconstituted a Peli2-KO cells caspase-1 cleavage is higher than in WT cells (possibly an artifact of overexpression).

Given the data from three experiments, the text does not appropriately represent the experimental data with respect to caspase-1 processing (“This is further supported by the observation that the re-introduction of WT Pellino2 could re-constitute the LPS/ATP-induced processing of pro-caspase-1 in Peli2-/- BMDMs to the same degree as displayed in WT cells whereas both the FHA and RING-like mutants of Pellino2 were ineffective in this regard (Fig. 5c, 5d).”).

Response: Firstly, to alleviate any concern of the Reviewer in relation to the reproducibility of these data, we have performed an additional 2 replicates of the

experiments (Figure 5c; n=4 and n=5; see page 2 of “Additional Data” file). Replicate (n=5) has now been included as Figure 5c in the revised manuscript. Furthermore, as requested by the Reviewer we have also conducted densitometry and statistical analysis across all 5 replicate experiments to quantitate levels of caspase 1 p20 and IL-1 β p17 in supernatants from WT and Peli2-KO cells (Figure 5d n=1-5; see page 2 of “Additional Data” file). For these analyses, given that the caspase 1 p20 and IL-1 β p17 blots were processed using the LiCor IR detection system, we performed direct quantitative analysis on the original IR scans of the blots (using the LiCor Western imaging software). The data unequivocally show reduced processing of caspase1 (p<0.001) and IL-1 β (p<0.0001) in Pellino2-deficient cells and these effects are rescued by re-constitution with wild type Pellino2 but not with forms with loss of function mutations in the FHA and RING-like domains. These data are now included as Figure 5d in the revised manuscript.

The data on IL1b-p17 from reconstituted Peli2-KO cells are convincing.

3. See 2. Caspase-1 processing data are still not convincing. IL-1b is OK.

Response: As described above we have now performed an additional 2 replicates of the experiments (see Figure 5c; n=4 and n=5). Furthermore, as requested by the Reviewer we have also conducted densitometry and statistical analysis across all 5 replicate experiments to quantitate levels of caspase 1 p20 in supernatants from WT and Peli2-KO cells (see Figure 5d n=1-5). The data clearly show reduced processing of caspase1 (p<0.001) in Pellino2-deficient cells.

4. While the data on the effects of MCC950 on the linkage-specific ubiquitination of NLRP3 are interesting, I do not see how these results add to the understanding of the function of Pellino2 in inflammasome activation or NLRP3 ubiquitination. Thus, and for conciseness sake, I suggest taking out these data altogether.

Response: By performing additional requested studies in the revised manuscript, we revealed increased complexity of the regulatory effects of MCC950 on ubiquitination of NLRP3. The value of including the data on MCC950 is the demonstration that K63-linked ubiquitination of NLRP3 is associated with activation of the pathway. Our model proposes that Signal 1 promotes K63-linked ubiquitination of NLRP3 as part of priming the NLRP3 inflammasome for activation. The NLRP3 activation phase is then associated with de-ubiquitination of NLRP3. Additional correlative support for the requirement of K63-linked ubiquitination of NLRP3 for its activation is provided by the specific NLRP3 inhibitor MCC950 demonstrating inhibitory effects on K63-linked ubiquitination of NLRP3. However, if helpful, we would be happy for the data to be removed for the sake of clarity and conciseness.

5. The study of Ordureau et al. (cited by the authors) shows that in the case of Pellino1, phosphorylation by IRAK4 clearly enhances Pellino1’s E3 ligase activity. I still believe that it would add mechanistical depth to test if IRAK1-dependent phosphorylation of Pellino2 is also relevant for Pellino2’s E3 activity and subsequent NLRP3 inflammasome activation,

independently of whether or not Pellino2 is directly phosphorylated by IRAK1 in this context.

Response: It is highly unlikely that IRAK1-induced phosphorylation and activation of Pellino2 is upstream of NLRP3 inflammasome activation since we have already shown enhanced interaction of Pellino2 with NLRP3 and increased ubiquitination and activation of NLRP3 in IRAK1-deficient cells. If helpful to the Reviewer, we can try to perform *in vitro* ubiquitination assays with recombinant Pellino2 in the absence and presence of recombinant IRAK1 to assess if IRAK1 can directly affect the E3 ubiquitin ligase activity of Pellino2. However, it is not clear how such studies could be directly integrated into NLRP3 activation or progress beyond our present understanding of the physiological relevance of IRAK1 to NLRP3 activation, as deduced from our studies using IRAK knockout cells.

6. OK

7. The changes to the text are OK. The study cited by the authors (Ordureau et al (2008) Biochem J. 409,43-52), however, does not show that IRAK1 is ubiquitinated by Pellino2. This would require an in vitro assay with purified components, overexpression in HEK293T cells does not prove a direct enzyme-substrate relationship.

Response: We are happy to clarify this in the revised manuscript with the inclusion of the following text at the end of the results section “*Whilst over-expression studies have previously shown Pellino2 to promote ubiquitination of IRAK1⁴⁷ in vitro ubiquitination assays with purified forms of Pellino2 and IRAK would be required to confirm that IRAK1 is a direct substrate for Pellino2*”.

8. OK

9. The schematic in the current form is not intuitively understandable. Especially the use of different styles and colors of arrows is confusing/not explained. What does e.g. a solid black arrow mean? ‘Activates’, ‘recruits’, ‘ubiquitinates’, ‘leads to’, etc.? Also it would maybe be a good idea to show two panels showing the course of events in WT and Pellino2-KO cells, respectively.

Response: We welcome this very helpful suggestion from the Reviewer and as recommended have now included a new schematic in Figure 9 in the revised manuscript that contains 2 panels to represent the course of events in WT and Pellino2 KO cells.

Minor comments:

a. OK

b. The b-actin band migrates around 42kDa and is therefore not a good control to assess transfer of proteins above 70kDa. A ponceau staining of the whole membrane would be more appropriate. Clearly there are issues with the ubiquitin blot in figure 8c (shades/white areas). Not publication quality.

Response: Given the Reviewer's comments that the blot is not publication quality we have included a replicate of this experiment with superior quality blots (Figure 8c; see page 4 of "Additional Data" file). We have also included 2 exposures of the anti-ubiq blot (IP IRAK) to clearly emphasise the reduced ubiquitination of IRAK in Pellino2-deficient cells. This is now included as Figure 8c in the revised manuscript.

Since we probe multiple proteins of different sizes by Western immunoblotting, we use β -actin as a common loading control since it migrates mid-range across the various proteins. We also find that Ponceau can decrease the subsequent quality of immunostained blots.

c. Figure 4d still says Capsase 11!

Response: This has been corrected in the revised manuscript.

d. Font sizes in the figures still vary widely (check for example legend of bar graphes vs. immunoblots in figure 2, 4, etc.; font size in fig. 2h vs. fig. 2m; figure 8)

Response: In the revised manuscript we have used font Arial10 in Figures, as recommended by Nature Communications.

e. OK

f. OK

Reviewer #2:

We welcome the support of this Reviewer for our work and its likely impact in the field; *"The authors have appropriately answered all my questions by performing new experiments or providing explanations. The paper will be received very well by the field."*

Reviewer #3:

The Reviewer acknowledges that *"The authors have performed a large number of additional experiments in order to address the concerns raised."*

However this Reviewer also highlights concerns; *"I find it disconcerting how in some cases the new data sets are dramatically different to those presented in the original manuscript (see below). However, even in the revised manuscript there remain data sets where the effect of Pell 2 deficiency on NLRP3 activity is not particularly convincing, or where two data sets show quite different results (e.g. point 5 below)."*

Response: We feel that these comments are harsh. Whilst experimental replicates show some biological variation, all of the data are in support of the overall

conclusions with none of the data contradicting our proposed model. However, in order to assuage the concerns of this Reviewer over the robustness of the data, we have included additional replicates and further optimized protocols below, together with statistical analysis to define any biological variation. Also, it should be highlighted that most of our conclusions are supported by a substantial body of multiple, independent and complementary strategies.

I address below the specific points raised by Reviewer#3.

- 1. Figure 2d/h. Unfortunately, the quality of these new blots is still quite poor. In particular, the supernatant IL-1 (two bands?) and caspase-1 bands are hazy/out of focus, and whether there is a real difference between WT and Pell 2 KO cells difficult to ascertain. Given the results are not black and white in WT and Pell 2 KOs, the gels should also have included an appropriate control (i.e. activation of an inflammasome that isn't impacted by Pell2 deficiency). Similarly, Figure 2m shows little, if any, difference between the caspase-1 and IL-1 cleavage fragments in the supernatants. Clear-cut data on this issue is essential for the authors to conclude that Pell 2 impacts NLRP3 activity (i.e. if clear demonstrable reductions in NLRP3 induced caspase-1 processing are not observed then this calls into question the manuscripts main conclusion). I also find it strange that if NLRP3 activation is impacted by the loss of Pell2, then why do WT and Pell2 KO cells secrete equivalent levels of IL-1 at low doses of nigericin, as their rebuttal data demonstrates?**

Response: Firstly we accept that the clarity of bands in samples from supernatants are not as defined as samples from cell lysates. However it should be noted that these supernatants are in serum-containing medium and it is known that high levels of serum proteins and high salt concentration of serum can lead to low molecular weight proteins losing their sharpness of migration towards the bottom of the gel (where processed forms of IL-1 β and caspase1 migrate) and this manifests in fuzzy/hazy like bands. This is characteristic of the nature of serum-containing samples and is also apparent in other published papers in the NLRP3 field. However, we are keen to fully address the concerns of the Reviewer and to this end we have performed replicates of these experiments in which we treated supernatant samples with a matrix that facilitates desalting and reduction in abundant serum proteins. These processed samples allow for more uniform SDS-PAGE and superior blots. These replicates are now included (see Fig. 2d (see Page 6 of “Additional Data” file), Fig. 2h (see Page 7 of “Additional Data” file) and 2m (see Page 8 of “Additional Data” file)) and clearly show reduced processing of caspase 1 and IL-1 β in Pellino2-deficient cells. As requested by the Reviewer these replicates also contain the stimulation control (Poly (dA:dT) confirming that AIM2 inflammasome activation is not affected by Pellino2 deficiency. These high quality blots (Fig. 2d, 2h and 2m) are now included in the revised manuscript. Furthermore, in relation to the original data and to assuage any concerns of reproducibility, we have included below densitometric analysis (by quantitation of original IR scans from LiCor system as described above) that

represents collated data of replicate experiments that confirms statistically significant reduction of caspase1 cleavage in Pellino2-deficient cells.

Densitometry analysis of Caspase-1 in medium (Sup) from WT and Pel12^{-/-} BMDMs stimulated with 100 ng/ml LPS for 3 h followed by 2.5mM ATP or 5mM Nigericin for 1 h. Data represents the mean +/- S.E.M of 3 independent experiments. *p<0.05, **p<0.01 (paired, two-tailed student's t-test).

Whilst these data, in their own standing, provide strong support for impairment of NLRP3 activation in Pellino2-deficient cells, the contention of the reviewer that ... “if clear demonstrable reductions in NLRP3 induced caspase-1 processing are not observed then this calls into question the manuscript's main conclusion” should be considered in the context that we have also supported this conclusion with 9 other independent approaches including:

- reduction of LPS/ATP-induced ASC oligomerization in Pellino2-deficient cells
- reduction of LPS/nigericin-induced ASC oligomerization in Pellino2-deficient cells
- reduced LPS/ATP–induced ASC speck formation in Pellino2-deficient cells
- reduced IL-1 β production in Pellino2-deficient cells in response to LPS/ATP or LPS/nigericin
- reduced IL-18 production in Pellino2-deficient cells in response to LPS/ATP or LPS/nigericin
- reduced pyroptosis in Pellino2-deficient cells in response to LPS/ATP or LPS/nigericin
- reduced IL-1 β and IL-18 production in Pellino2-deficient cells in response to various Gram-negative bacteria
- reduction of levels of IL-1 β and lethality in Pellino2-deficient mice in response to *in vivo* administration of LPS
- reduction of levels of IL-1 β and IL-18 in Pellino2-deficient mice in response to infection with *Pseudomonas*.

The Reviewer also comments that “*I also find it strange that if NLRP3 activation is impacted by the loss of Pell2, then why do WT and Pell2 KO cells secrete equivalent levels of IL-1 at low doses of nigericin, as their rebuttal data demonstrates?*” Firstly, virtually all papers in the NLRP3 field use the same concentrations as we used in the original submission. However we were keen to address all comments and so after performing the dose-dependent experiments for the revised manuscript it is surprising that the Reviewer selectively focuses on a low dose of nigericin in which minimal levels of IL-1 β are produced without commenting that inhibition is observed at all 5 other doses of nigericin. Furthermore, we also see strong inhibition at all effective doses of ATP.

2. Figure 2l/m. I still fail to see why the western blots don’t detect cleaved IL-1/casp1 with LPS stimulation alone given that the levels of IL-1 detected by ELISA are almost on par (and were identical in the original manuscript) with the LPS/ATP, LPS/Nigericin stimulations.

Response: Firstly, the levels of IL-1 β that are produced by THP1 cells are much lower than the amounts produced by primary macrophages in the rest of the study. Indeed as part of submitting the revised manuscript, we tried to optimize the experimental model to maximize the secretion of IL-1 β and so extended the LPS time treatments from 3 hours to 6 hours. This allowed for some improvement in the expression levels of IL-1 β and this was captured in the ELISA analysis that showed IL-1 β expression after 6 hours of priming with LPS. However the western blots represented a LPS priming time of 3 hour, at which time processed IL-1 β and caspase1 are not apparent in response to stimulation of cells with LPS alone. We have now included new data (Fig 2m) in the revised manuscript of western blots showing processed IL-1 β and caspase1 (including with LPS alone) at 6 hours of LPS priming (to coincide with 6h LPS priming time for ELISA analysis in Fig. 2l). Furthermore, to clarify any confusion from the previous submission, I also include data in the “Additional Data” file (see Page 8; Fig. 2m) that directly compares LPS priming times of 3 hours and 6 hours and it is clear that LPS alone (6h) promotes processing of IL-1 β and caspase1 (in agreement with ELISA data) whereas LPS alone (3h) fails to induce detectable levels of these processed forms.

3. Original point 9. Caspase-11 activates NLRP3 via potassium efflux, which is how nigericin and ATP also activate NLRP3. This data actually better fits with the western blots and low dose nigericin treatments suggesting Pell2 loss results in normal levels of NLRP3 activity in several situations.

Response: In contrast to the Reviewers contention and as demonstrated in the revised manuscript, it is clear that the Western blots clearly show impaired NLRP3 activation in Pellino2-deficient cells. In addition, the Reviewer again focuses on a low dose of nigericin at which minimal levels of IL-1 β are produced whilst not acknowledging that all 5 other doses of nigericin show suppressed NLRP3 activity. We also strongly

contend that the Reviewer's comments that "*Pell2 loss results in normal levels of NLRP3 activity in several situations.*" is inaccurate feel that this statement does not reflect our data.

Furthermore, whilst caspase 11 activates NLRP3 via potassium efflux, the precise molecular events leading to this have not been fully elucidated and may differ depending on stimuli. Whilst there was no effect of Pellino2 deficiency on non-canonical inflammasome activation in response to cytosolic LPS or Cholera toxin, we showed that the loss of Pellino2 impaired secretion of IL-1 β that is induced by bacteria such as *E. coli* or *C. rodentium*. This suggests a potential role of Pellino2 under certain conditions involving caspase11-driven activation of NLRP3.

4. Figure 5C. It is very difficult to know which data set to believe. The original blot showing normal caspase-1 processing and secretion in Peli2 KO cells, or the new blot in the revised manuscript showing undetectable caspase-1 processing and secretion? The contradictory nature of these two data sets (and others, such as the original Figure 7f versus revised 7g) calls into question how consistent some of the results are.

Response: We feel that these comments do not fairly represent our data. Firstly, the original blot did not show "*normal Caspase-1 processing and secretion in Peli2 KO cells.*" and instead the cells showed clearly reduced levels of processed caspase1. We accept that the newer blots showed more reduction of processed caspase1 in Pellino2-deficient cells. Whilst there may be variation in the magnitude of this difference, this represents variation across biological replicates and not contradictory data. However to assuage any remaining concerns of the Reviewer on the robustness and reproducibility of these data, we have performed an additional 2 replicates of the experiments, including the de-salting of supernatant samples to facilitate higher quality blots (see Figure 5c; n=4 and n=5; page 2 of "Additional Data" file). Replicate (n=5) has now been included as Figure 5c in the revised manuscript. Furthermore, we have also conducted densitometry and statistical analysis across all 5 replicate experiments to quantitate levels of caspase 1 p20 and IL-1 β p17 in supernatants from WT and Peli2-KO cells (Figure 5d n=1-5; see page 2 of "Additional Data" file). The data unequivocally show reduced processing of caspase1 ($p < 0.001$) and IL-1 β ($p < 0.0001$) in Pellino2-deficient cells and these effects are rescued by re-constitution with wild type Pellino2 but not with forms with loss of function mutations in the FHA and RING-like domains. These data are now included as Figure 5d in the revised manuscript.

It should also be noted that the role for Pellino2 in NLRP3 activation in the re-constitution studies are also corroborated by independently demonstrating reduced ASC oligomerization in Pellino2-deficient cells that can be re-constituted by re-introduction of exogenous Pellino2 but not forms with loss of function mutations in their FHA and RING domains (Figure 6).

We also contend the Reviewers comments that Figure 7f and 7g are contradictory. This is not the case. The revised Figure 7g represents a more detailed kinetic study (as

requested by Reviewers) with both studies showing time-dependent interaction of Pellino2 with NLRP3.

5. Figure 6b-d. I find it concerning that one experiment with LPS/ATP shows a moderate reduction in ASC crosslinking upon Pell2 loss (b) while a second experiment shows a complete loss in ASC cross linking (c).

Response: As indicated above, there is some variation in relation to the magnitude of the loss of the response in the Pellino2-deficient but in all cases there is full agreement that activation is impaired with Pellino2 deficiency. Also it should be noted that reduced ASC oligomerisation in Pellino2 KO cells is fully corroborated by reduced ASC formation (see Figure 6a), again highlighting our efforts to confirm our conclusions with independent and complementary approaches.

6. The authors show Pell2 dependent ubiquitination of NLRP3, and write that “These data are consistent with a model in which LPS promotes transient K63-linked ubiquitination of NLRP3 that is associated with NLRP3 activation....”. I’m not sure this statement is warranted because LPS alone does not activate NLRP3. In addition, the authors data suggest that LPS triggers K63-ubiquitination of NLRP3 in 1.5hrs and that this completely disappears by 3 hrs. Therefore, it is very much a black hole as to what this NLRP3 ubiquitination is actually doing without experiments to define the E3 ligase responsible, or the lysine's modified and what happens to NLRP3 activity if these sites are mutated.

Response: We are happy to include clarification of our intended meaning in the revised manuscript. Whilst LPS alone doesn’t activate NLRP3, it does play an important role as part of the priming phase by regulating NLRP3 (e.g. expression and post-translational modification) to be receptive to a second signal that can promote its subsequent activation. We propose LPS-induced K63-linked ubiquitination of NLRP3 to be an early modification that may prime for later activation by a second signal and on page 11, line 14 of revised manuscript we now include the text “*Thus, LPS promotes transient K63-linked ubiquitination of NLRP3, that may prime NLRP3 for subsequent activation..*”.

We agree with the Reviewer that the molecular detail underlying ubiquitination of NLRP3 is of great interest but this goes way beyond the scope of the present study. We would like to address this over a more prolonged time frame but the challenge for these studies is very significant given that different types of polyubiquitin chains can be attached to NLRP3 coupled to NLRP3 possessing >70 lysine residues that are potential ubiquitin-acceptor sites.

7. Figure 8d. I still find it unclear how IRAK1 can be immunopurified from cells with almost undetectable IRAK1 levels. The authors state that “...IRAK...modified forms will not be visible at the region corresponding to native IRAK1 but will still be subject to immunoprecipitation and detection of interacting proteins such as NLRP3.” The IRAK1 doublet detected in the IP and the input blot are running at the same molecular weight (70-80kDa) implying that they are the same form that is almost completely degraded upon

LPS treatment. The 100 kDa IRAK1 form that doesn't appear to be degraded (e.g. Figure 8e, 8i) and is therefore arguably more likely to be available for complexing and IP doesn't actually appear to be detected in the IRAK1 IP samples (8e).

Response: Firstly, earlier studies by Ordureau et al (2008: Biochem. J. (409:43) used overexpression studies to demonstrate that Pellino2 can promote ubiquitination of IRAK-1 in cells. This manifested (in Western blots) as multiple faint bands of IRAK across a range of electrophoretic mobilities (due to varying levels of ubiquitination) with the levels of unmodified IRAK1 being heavily depleted. This supports the contention that strong reduction in levels of native IRAK does not mean that IRAK is absent from the cells but instead may be distributed across modified forms that can be immunopurified from cells.

Secondly, we have used 2 independent approaches to demonstrate that we can detect ubiquitinated forms of IRAK-1 at times post LPS stimulation when levels of native IRAK are greatly reduced. These include anti-ubiquitin immunoblotting of anti-IRAK immunoprecipitations (Fig. 8c) and also anti-IRAK1 blots of TUBE purified material (Fig. 8d). These independent approaches clearly show that IRAK can be immunopurified from LPS treated cells.

Finally, the Reviewer queries Figure 8e in that the 100kDa IRAK band doesn't appear to be detected in the IP samples. We welcome the opportunity to clarify this confusion. The band that is depleted by LPS in this panel is the 100 kDa band. In previously cropping the panel for figure design, we had inadvertently cropped too much of the panel and the lower 70kda bands were not visible. I have uncropped this region of the blot and this is included in the figure below and as Fig.8e in the revised manuscript. I trust this clarified any confusion. I apologise for this error and for not noting it during our original proofreading.

I hope that we have addressed all of the issues articulated by the Reviewers and have demonstrated our willingness to fully satisfy all concerns. As part of this appeal we have also attached all uncropped blots of new data (see pages 3-8 of “Additional Data” file) to provide a sense of the overall quality of our data.

Thank you again for considering our work.

Kind Regards

Paul

Prof. Paul Moynagh

Corresponding Author

REVIEWERS' COMMENTS:

Reviewer #1 (Remarks to the Author):

The authors have carefully revised the manuscript according to my previous comments. Most questions were appropriately answered although my previous major point 5 and the minor point b are still not fully addressed.

Reviewer #3 (Remarks to the Author):

I very much appreciate the extent the authors have gone to to allay my concerns in their rebuttal letter. The robustness and quality of the data in the manuscript now justify the conclusions and the manuscripts publication; I hope the it is well received by the inflammasome field.

Prof. Paul Moynagh
Department of Biology
Maynooth University
Co. Kildare
Ireland
Tel: 0035317086105
Email: Paul.Moynagh@mu.ie

23rd February 2018

Dear Editor,

Please find submitted a revised Manuscript entitled " **The E3 ubiquitin ligase Pellino2 mediates priming of the NLRP3 inflammasome.**" for consideration for final publication in *Nature Communications*. We welcome the positive responses of Reviewers #1 and #3 in supporting publication of the manuscript. Specific responses to individual Reviewer comments are included below

Reviewer #1 (Remarks to the Author):

The authors have carefully revised the manuscript according to my previous comments. Most questions were appropriately answered although my previous major point 5 and the minor point b are still not fully addressed.

Response: Text edits are included as tracked changes in the revised manuscript to address these points. These text changes have been previously approved by the editor.

Reviewer #3 (Remarks to the Author):

I very much appreciate the extent the authors have gone to to allay my concerns in their rebuttal letter. The robustness and quality of the data in the manuscript now justify the conclusions and the manuscripts publication; I hope the it is well received by the inflammasome field.

Response: We welcome the positive response and support of Reviewer#3.

I hope that the revised manuscript is acceptable for final publication in *Nature Communications*.

Kind Regards

Prof. Paul Moynagh
Corresponding Author